# LDL receptor-related protein 5 selectively transports unesterified polyunsaturated fatty acids to intracellular compartments

Wenwen Tang [1,2,7] ✉, Yi Luan[1,2,7], Qianying Yuan[1,2,7], Ao Li[1,2], Song Chen [2], Stanley Menacherry[3], Lawrence Young[4,5,6] & Dianqing Wu [1,2] ✉

Polyunsaturated fatty acids (PUFAs), which cannot be synthesized by animals and must be supplied from the diet, have been strongly associated with human health. However, the mechanisms for their accretion remain poorly understood. Here, we show that LDL receptor-related protein 5 (LRP5), but not its homolog LRP6, selectively transports unesterified PUFAs into a number of cell types. The LDLa ligand-binding repeats of LRP5 directly bind to PUFAs and are required and sufficient for PUFA transport. In contrast to the known PUFA transporters Mfsd2a, CD36 and FATP2, LRP5 transports unesterified PUFAs via internalization to intracellular compartments including lysosomes, and n-3 PUFAs depend on this transport mechanism to inhibit mTORC1. This LRP5-mediated PUFA transport mechanism suppresses extracellular trap formation in neutrophils and protects mice from myocardial injury during ischemia-reperfusion. Thus, this study reveals a biologically important mechanism for unesterified PUFA transport to intracellular compartments.

Fatty acids are classified into three major classes: saturated fatty acids (no double bonds), monounsaturated fatty acids (a single double bond), and polyunsaturated fatty acids (PUFAs) with multiple double bonds. Depending on the methyl terminal position of the first double bond, PUFAs are further divided into n-3 (or ω3) and n-6 (or ω6) PUFAs. The n-3 PUFAs include α-linolenic acid (ALA), eicosapentaenoic acid (EPA), and docosahexaenoic acid (DHA), whereas the n-6 PUFAs include linoleic acid (LA) and arachidonic acid (ARA). PUFAs are essential fatty acids, which cannot be synthesized by humans, mice, and other mammals and thus must be supplied in the diet[1,2]. PUFAs, particularly n-3 PUFAs, are generally considered to be anti-inflammatory and have beneficial effects on human health and particularly cardiovascular health including potential protection from myocardial ischemia-reperfusion injury[1–9]. While the molecular mechanisms accounting for these effects of PUFAs remain under-investigated, PUFA-derived bioactive metabolites regulate various aspects of inflammatory processes[10–12]. In addition, PUFAs can also signal through cell surface G protein-coupled receptors including GPR40 and GPR120[13]. Moreover, studies showed that n-3 PUFAs inhibited mTORC1 signaling[14–18]. The mTORC1 signaling pathways regulate numerous important cellular processes including stimulation of protein synthesis primarily through phosphorylation of p70S6 kinase 1 (S6K), S6 protein, and eIF4E Binding Protein (4EBP)[19].

Although fatty acids can passively diffuse across the cell plasma membrane, cell surface transporters, including the CD36 class molecules and fatty acid transporter proteins (FATPs), facilitate the uptake of fatty acids including PUFAs[20,21]. The specific functions of PUFAs transported by CD36 and FATPs, except for FATP2, remain poorly defined. FATP2 is highly expressed in polymorphonuclear myeloid-derived suppressor cells in tumor-bearing mice and transports ARA into these cells to enhance their immunosuppressive activities[22]. However, it is expressed at a very low level and has a negligible role in

[1]Vascular Biology and Therapeutic Program, Yale University School of Medicine, New Haven, CT 06520, USA. [2]Department of Pharmacology, Yale University School of Medicine, New Haven, CT 06520, USA. [3]Gateway Community College, New Haven, CT 06510, USA. [4]Cardiovascular Research Center, Yale University School of Medicine, New Haven, CT 06520, USA. [5]Department of Internal Medicine (Cardiovascular Medicine), Yale University School of Medicine, New Haven, CT 06520, USA. [6]Department of Cellular and Molecular Physiology, Yale University School of Medicine, New Haven, CT 06520, USA. [7]These authors contributed equally: Wenwen Tang, Yi Luan, Qianying Yuan. ✉e-mail: wenwen.tang@yale.edu; dan.wu@yale.edu

PUFA transport in normal neutrophils[22]. Another known transporter for PUFAs is Mfsd2a, which exclusively transports esterified, but not non-esterified, DHA and is important for PUFA accretion into the central nervous system[23,24]. Thus, there is still a general lack of knowledge on the mechanisms for PUFA accretion and the specific function of PUFAs in most tissues and cell types.

LRP5 is a single transmembrane protein and shares close amino acid sequence homology with LRP6. Its extracellular domain contains four YWTD repeat domains and three LDL receptor Class A (LDLa) or ligand-binding repeats[25]. The YWTD repeat domains are involved in interactions with Wnt proteins and Wnt antagonists including DKK and sclerostin[26], whereas no specific function has been linked to the LDLa repeats. These LDLa repeats are homologous with those in the LDL receptor that mediate lipoprotein binding[27]. LRP5 can function as a Wnt co-receptor in β-catenin stabilization via the binding of its intracellular domain to axin[28]. Studies, particularly mouse genetic studies, have further confirmed that LRP5 contributes to Wnt signaling, however, LRP6 seems to be the dominant co-receptor in Wnt-β-catenin signaling in most cases[29–33]. On the other hand, studies of LRP5 KO mice also reveal that some aspects of the LRP5 KO phenotype might not be readily connected to perturbation of Wnt-β-catenin signaling[34–39]. In this study, we identified LRP5 as being the transporter selective for unesterified PUFA and showed that LRP5 is required for mTORC1 suppression by unesterified n-3 PUFAs, which is important for neutrophil function.

## Results

### Neutrophil LRP5 protects mice from myocardial ischemia-reperfusion injury

Wnt-β-catenin signaling was previously implicated in neutrophil biology in the context of tumor immunity[40], and global LRP5-deficiency was shown to increase the extent of injury in a mouse model of myocardial infarction[41]. Thus, we investigated the roles of myeloid LRP5 and 6 in modulating injury during myocardial ischemia-reperfusion, a condition in which neutrophils play an important role[42,43]. Myeloid-specific LRP5 (*Lrp5*[M]) or LRP6 (*Lrp6*[M]) knockout (KO) mice, driven by Lyz2-Cre, were subjected to left coronary artery ligation and reperfusion. The *Lrp5*[M] KO mice sustained significantly greater myocardial ischemia-reperfusion injury, as indicated by larger areas of myocardial infarction (normalized to the ischemic risk region produced by coronary artery ligation) and greater impairment in contractile function (as indicated by lower left ventricular ejection fractions) than those of the WT littermates (Fig. 1a, c). In contrast, myeloid-specific LRP6-deficiency mice had no significant differences in myocardial ischemia-reperfusion injury compared to their WT controls (Fig.1b, c). LRP5-deficiency did not significantly alter the circulating neutrophil abundance prior to or after ischemia-reperfusion (Supplementary Fig. 1a). However, the *Lrp5*[M] mice had moderately increased neutrophils, but not monocytes, in their hearts compared to WT, after ischemia-reperfusion (Fig. 1d and Supplementary Fig. 1b, c). Moreover, there was significantly increased co-staining of MPO (a neutrophil marker) and citrullinated histone H3 (a NET marker) in hearts of *Lrp5*[M] KO mice vs. WT controls after ischemia-reperfusion (Fig. 1e, f), suggesting that myeloid-specific LRP5-deficiency increases neutrophil NET formation and probably activation. Consistent with this in vivo observation of the increased formation of NETs in the LRP5-null hearts during ischemia-reperfusion, isolated LRP5-deficient neutrophils also had more citrullinated H3 histone than WT neutrophils upon stimulation (Fig. 1g) and produced more extracellular DNA than the WT cells (Fig. 1h).

These results, together with the knowledge of the importance of neutrophils in myocardial ischemia-reperfusion injury, suggest that neutrophil LRP5 may be responsible for the observed phenotypes. Thus, we tested and found that Mrp8-Cre-driven LRP5 (*Lrp5*[N]) KO mice also had increased myocardial ischemia-reperfusion injury and elevated neutrophil NET formation (Fig. 2a and Supplementary Fig. 2a, b).

Since Mrp8-Cre is more specific than Lyz2-Cre for neutrophils[44], these results indicate that neutrophil LRP5 specifically protects hearts from ischemia-reperfusion injury.

### LRP5 is important for PUFA accretion in neutrophils

To understand the downstream signaling mechanism for LRP5 in neutrophils, we first examined the role of LRP5 in Wnt-induced β-catenin stabilization in neutrophils. We found that LRP5 deficiency had little effect on Wnt3a-stimulated β-catenin accumulation in neutrophils (Supplementary Fig. 2c). In contrast, LRP6-deficiency abrogated Wnt3a-stimulated β-catenin accumulation in neutrophils (Supplementary Fig. 2d). This result indicates that LRP6, rather than LRP5, mediates Wnt-β-catenin signaling in neutrophils. To identify the mechanism by which LRP5 functions in neutrophils, we performed shotgun lipidomic, transcriptomic, and metabolomic analyses of LRP5-null neutrophils in comparison to the WT cells. The shotgun lipidomic analysis revealed marked differences in the contents of glycerolipid species between LRP5-null and WT neutrophils (Supplementary Data 1). Most noticeably, all of the triglycerides containing more than 5 double bonds and near sixty percent of the phospholipids with one of its fatty acid chains containing 5 double bonds identified by the analysis showed significant reductions in LRP5-null neutrophils compared to WT (Fig. 2b and Supplementary Fig. 2f). The dominant type of 5 double-bond fatty acids in the phospholipid group is 20:5 and likely EPA. In addition, twenty percent of the phospholipids with one of their fatty acid chains containing 4 double-bonds showed significant reductions in LRP5-null neutrophils (Supplementary Fig. 2h). By contrast, none of the phospholipids containing the saturated fatty acid chains identified by the analysis showed significant reductions (Fig. 2b and Supplementary Fig. 2g). A few of other phospholipids identified by the shotgun lipidomics also showed significant changes (Supplementary Fig. 2h). These shotgun lipidomic results suggest that LRP5-deficiency appears to reduce PUFAs contents in neutrophils. Targeted LC-MS analysis results showing reduced contents of PC, PE and PA species containing PUFAs in LRP5-deficient neutrophils compared to WT neutrophils (Supplementary Fig. 2e) further corroborates this possibility.

To further test the hypothesis that LRP5-deficiency significantly reduced PUFA accretion in neutrophils, we examined the role of LRP5 on PUFA uptake. We found that LRP5-deficiency resulted in a significant reduction in the uptake of [14]C-EPA into neutrophils compared to the WT cells in the presence or absence of the fatty acid carrier protein bovine serum albumin (Fig. 2d). This result was further confirmed by detecting less deuterated EPA in neutrophils that were isolated from *Lrp5*[N] KO compared to those from control WT mice fed on the essential fatty acid-free diet and cultured with deuterated EPA (Fig. 2e). LRP5-deficiency also led to a reduction in [14]C-ARA uptake by neutrophils (Fig. 2f). In contrast, LRP5-deficiency had no effect on [14]C-oleic acid (OL, a mono-unsaturated fatty acid) uptake (Fig. 2f). These results together indicate that LRP5 is a PUFA transporter in neutrophils.

To determine if LRP5-transported PUFAs play an important role in neutrophil function in vivo, we examined the effects of PUFA-deprivation on myocardial ischemia-reperfusion injury. Because PUFAs cannot be synthesized and must be supplied from the diet, mice fed an essential fatty acid-free diet are deprived of PUFAs. In contrast to the previous findings in mice on a normal diet, the essential fatty acid-free diet abrogated the differences in myocardial ischemia-reperfusion injury and NET formation between the neutrophil-specific LRP5 KO and control WT mice (Fig. 2a and Supplementary Fig. 2b). Our lipidomic and target LC-MS analyses showed that the EFA-free diet led to substantial decreases preferentially in PUFA-containing lipids within neutrophils without significant alterations in the levels of saturated or monounsaturated FA-containing lipids (Fig. 2c, Supplementary Fig. 2i, j and Supplementary Data 2). In addition, we also found that there was a significant reduction in EPA levels in the plasma of

mice on the EFA-free diet compared to those on the normal diet (Supplementary Fig. 2k). Thus, the essential fatty acid-free diet replicated the effects of neutrophil-specific LRP5 KO on increasing NET formation and exacerbating myocardial ischemia-reperfusion injury in the WT mice and had no additive effects in the LRP5 KO mice. These findings suggest that neutrophil PUFA depletion may account for the adverse phenotype of LRP5 KO mice during myocardial ischemia-reperfusion.

## LRP5 is important for PUFA accretion in many cell types

We went on to examine if LRP5 transports PUFA in other cell types. We performed the $^{14}$C-EPA and $^{14}$C-ARA uptake assay using NK cells isolated from Ncr1-Cre LRP5$^{fl/fl}$, macrophages from Lyz-Cre LRP5$^{fl/fl}$ mice, T lymphocytes, B lymphocytes, fibroblasts, and endothelial cells from Rosa26-CreER2 LRP5$^{fl/fl}$ mice treated with tamoxifen, together with their corresponding LRP5 WT control cells. LRP5 deficiency resulted in significant reductions in $^{14}$C-EPA and $^{14}$C-ARA uptake in NK cells,

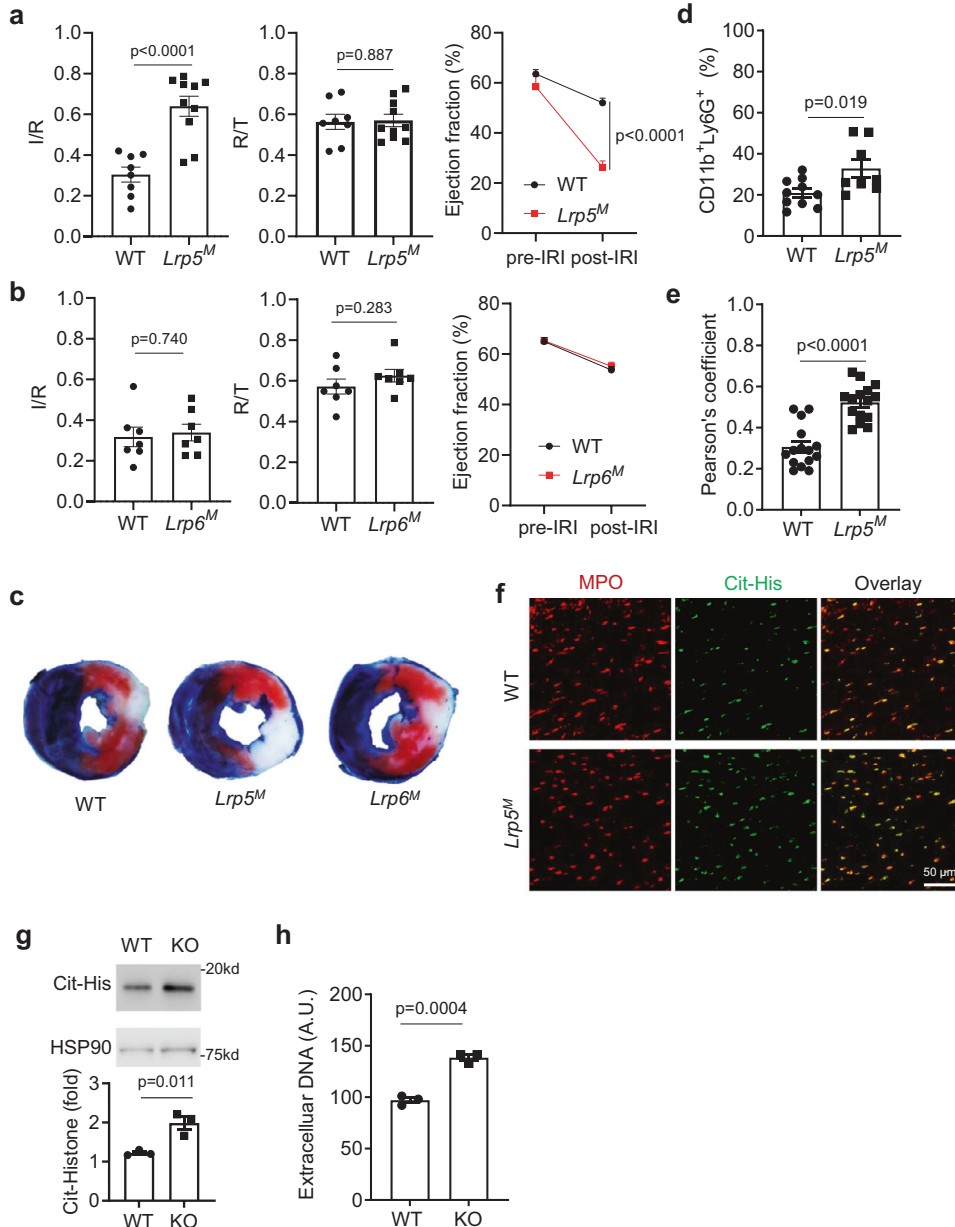

**Fig. 1 | Myeloid-specific LRP5, but not LRP6, KO, leads to increased myocardial ischemia-reperfusion injury and NET formation.** Mice lacking LRP5 (*Lrp5$^{M}$*, (**a**)) or LRP6 (*Lrp6$^{M}$*, (**b**)) in myeloid cells and their corresponding wildtype (WT) littermate control mice were subjected to myocardial ischemia-reperfusion injury. Infarction area (I) normalized against risk area (R) and Ejection Fractions before and after myocardial ischemia-reperfusion injury are shown. The ratios of risk area (R) to total area (T) are also shown. Each datum point represents one mouse (n = 8 for WT and n = 10 for *Lrp5$^{M}$* in (**a**). n = 7 for WT and n = 7 for *Lrp6$^{M}$* in (**b**)). Data were combined from three independent experiments. **c** Representative heart section images stained with triphenyl tetrazolium chloride from mice subjected to myocardial ischemia-reperfusion injury for (**a**, **b**). **d** Neutrophil presence in injured hearts determined by flowcytometry. Each datum point represents one mouse

(n = 10 for WT and n = 8 for *Lrp5$^{M}$*). Data were combined from three independent experiments. **e** Pearson's coefficients of co-localization of MPO and citrullinated histone (c-His) in the injured heart sections. Each datum point is one section. Three independent sections per mouse and five mice per group were analyzed. **f** Representative images of heart sections stained with anti-MPO and anti-citrullinated histone (Cit-His). **g**, **h** LRP5-deificient (KO) and wildtype (WT) neutrophils were subjected to Western analysis to examine citrullinated histone or ELISA to detect extracellular DNA. Each datum point represents one mouse (n = 3 for WT and n = 3 for KO). Data were combined from three independent experiments. Data in this figure are all presented as mean ± sem with *p* values (Student's *t*-test, Two-tailed, unpaired).

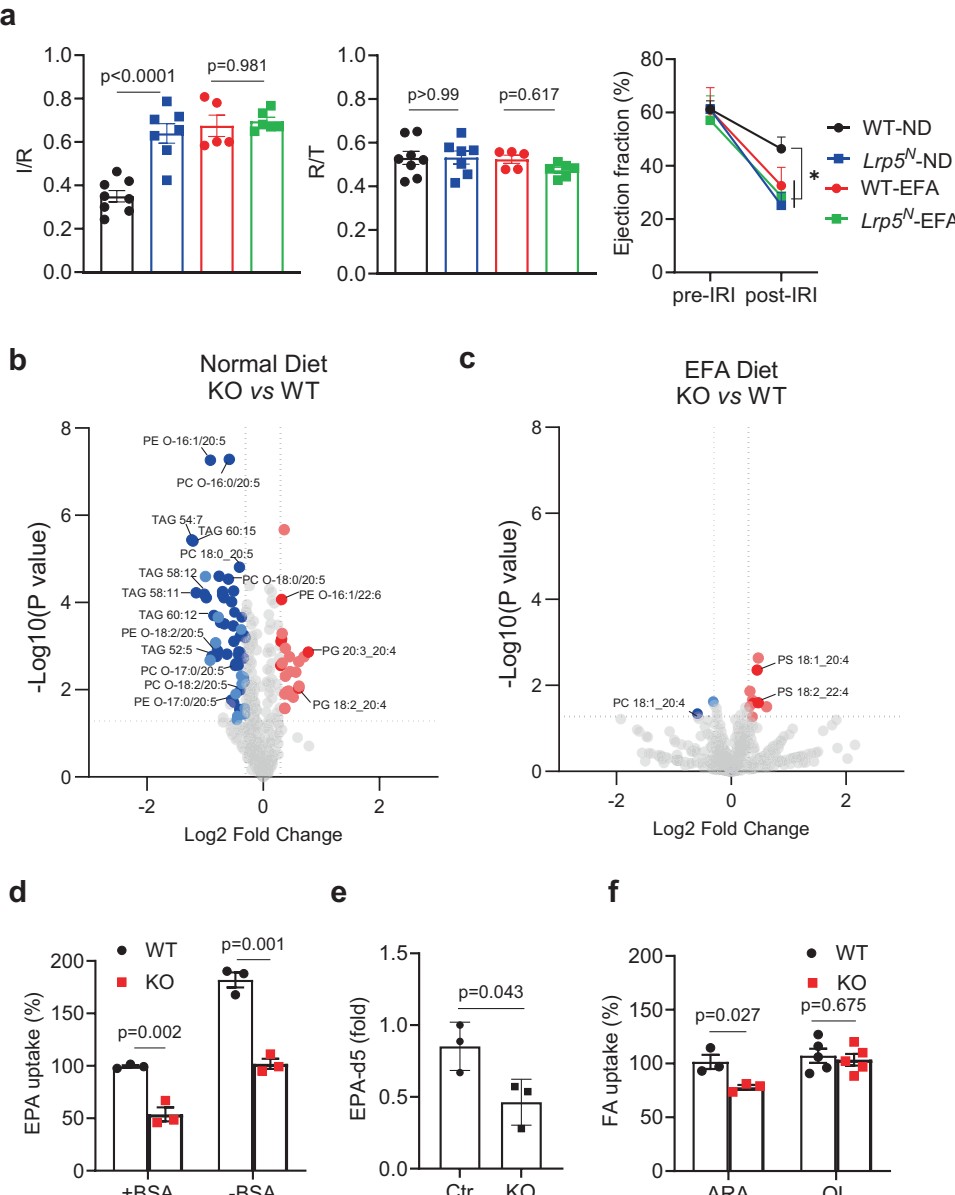

**Fig. 2 | LRP5-deficiency reduces PUFA accretion in neutrophils. a** Mice lacking LRP5 in neutrophils (*Lrp5^N*) and their corresponding wildtype (WT) littermate control mice on the normal diet (ND) or essential fatty acid-free diets (EFA) were subjected to myocardial ischemia-reperfusion injury. Infarction area (I) normalized against risk area (R) and Ejection Fractions before and after myocardial ischemia-reperfusion injury are shown. Each datum point represents one mouse (*n* = 8 for WT-ND(black circle), *n* = 7 for *Lrp5^N*-ND(blue square), *n* = 5 for WT-EFA(red circle) and *n* = 6 for *Lrp5^N*-EFA(green square)). Data were combined from three independent experiments. Data are presented as mean ± sem (One-way ANOVA for I/R and R/T, Two-way ANOVA for Ejection fraction, *p* < 0.05). Volcano plots showing lipid species that are significantly (*p* < 0.05, Log₂Fold > 0.3 or < −0.3, Student's *t*-test, Two-tailed, unpaired) increased (red) and decreased (blue)

in LRP5-deficient neutrophils compared to corresponding WT cells from mice on normal diet (**b**) or EFA-free diet (**c**). Dark blue and dark red points are lipids containing PUFAs; light blue and light red points are lipids not containing PUFAs. The complete dataset is shown in Supplementary Data 1 and Data 2. **d, f** Neutrophils isolated from *Lrp5^N* (KO) or WT mice fed on the essential fatty acid-free diet were incubated with ¹⁴C-EPA in the presence and absence of BSA or ¹⁴C-ARA or ¹⁴C-oleic acid (OL) with BSA. The uptake by WT cells is taken as 100% (*n* = 3 per group in (**d**) and ARA group in (**f**), *n* = 5 per group in OL group in (**f**)). **e** Targeted LC-MS analysis of deuterated EPA in neutrophils (*n* = 3 per group). Data in (**d**–**f**) are presented as mean ± sem with *p* values (Student's *t*-test, Two-tailed, unpaired).

macrophages, endothelial cells, and fibroblast cells (Fig. 3a), but not in splenic naïve T lymphocytes or B lymphocytes (Fig. 3a). We also observed a decrease in ¹⁴C-EPA uptake by LRP5 KO HEK293T cells created by CRISPR/Cas9 in comparison with WT HEK293T cells (Fig. 3b). Of note, naïve splenic T cells and B cells appeared to have low expression of the LRP5 protein (Supplementary Fig. 2l). Thus, many other cell types other than neutrophils, if not all, also utilize LRP5 for the uptake of PUFAs.

## LRP5 LDLa repeat domain binds to PUFAs

Because the LDLa repeats in the extracellular domain of LRP5 (Supplementary Fig. 3a) are homologous with those in the LDL receptor that mediate lipoprotein binding and transport[27], we suspected that these repeats might be involved in PUFA transport. Indeed, when a LRP5 mutant lacking the LDLa repeats was expressed in the LRP5 KO HEK293T cells, there were no increases in ¹⁴C-EPA uptake (Supplementary Fig. 3b). In contrast, overexpression of WT LRP5 increased

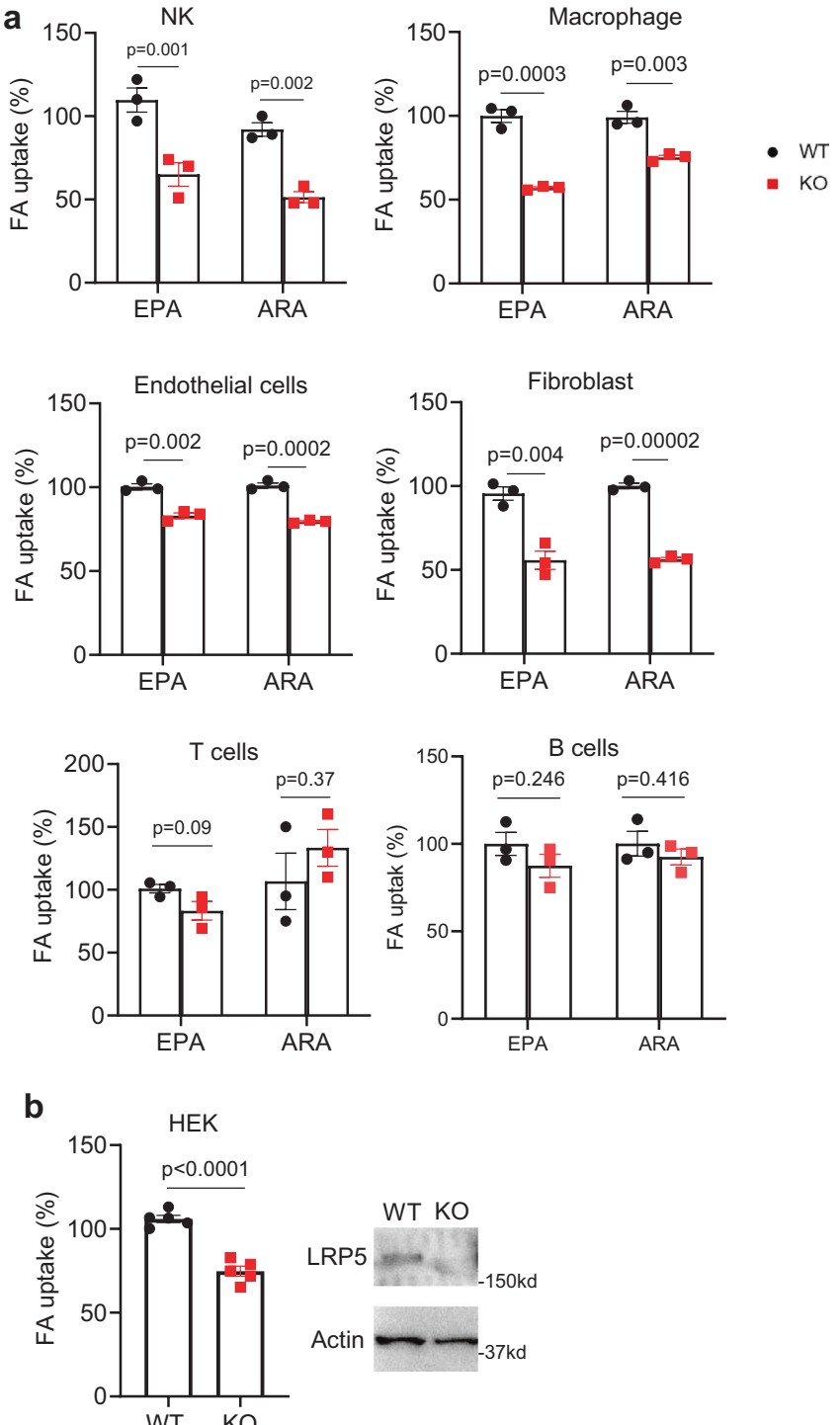

**Fig. 3 | LRP5 deficiency reduces PUFAs uptake in various cell types. a** [14]C-EPA and [14]C-ARA uptakes were determined using splenic NK cells isolated from NCR-Cre LRP5 KO and WT littermates, peritoneal macrophages isolated from *Lrp5[M]* and littermate WT mice, naïve splenic T lymphocytes, B lymphocytes, pulmonary fibroblast cells, and pulmonary endothelial cells prepared from Rosa-CreER2-driven LRP5 conditional KO and littermate WT treated with tamoxifen (*n* = 3 per group). **b** [14]C-EPA uptake was determined using LRP5 KO HEK293T cells and WT cells (*n* = 5 per group). LRP5 KO HEK293T cells were generated by CRISPR/Case9 and validated by Western analysis. Data in this figure are all presented as mean ± sem with *p* values (Student's *t*-test, Two-tailed, unpaired).

[14]C-EPA uptake (Supplementary Fig. 3b). In addition, the deletion of the four YWTD repeat domains at the N-terminal of LRP5 (LRP5-dN) didn't affect the LRP5-mediated [14]C-EPA uptake (Supplementary Fig. 3b). These data together suggest that the LDLa repeat domain, which is proximal to the transmembrane domain, is required and sufficient for the LRP5-mediated uptake of EPA.

To determine if these repeats are involved in binding to PUFAs, we prepared conditioned medium (CM) from cells expressing a secreted form of LRP5 LDLa repeats fused with human IgG1 Fc region, designated as LDLa-Fc, and the control CM containing a secreted form of Fc (Supplementary Fig. 3a, c). These CMs were added to a 96-well plate coated with BSA or BSA in the presence of one fatty acid

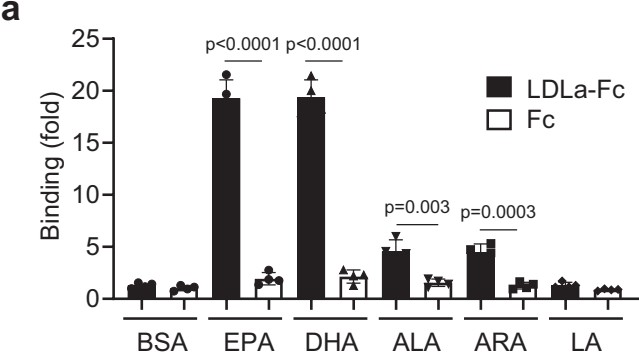

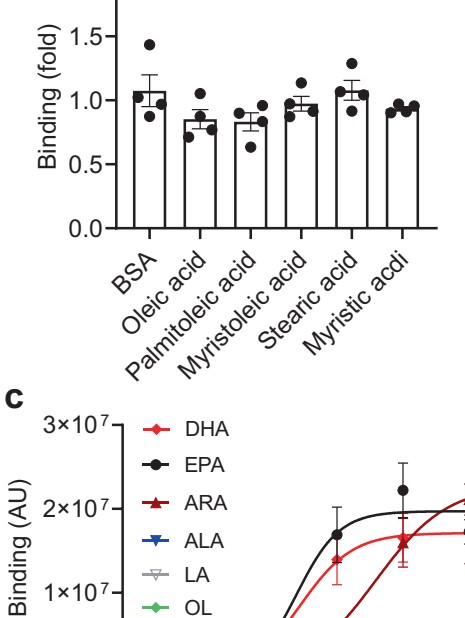

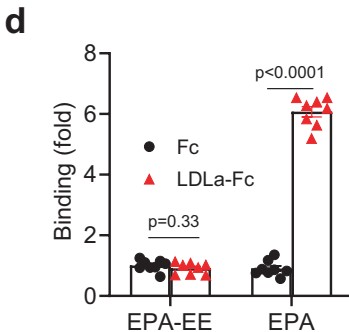

**Fig. 4 | LRP5 LDLa repeats bind to PUFAs. a, b** Binding of LRP5 LDLa repeats to fatty acids was determined by coating the plate with BSA or BSA plus the indicated fatty acids, followed by incubation with conditioned medium from cells expressing Fc or LRP5 LDLa-Fc. The binding of Fc to BSA is set as 1 ($n = 4$ per group). **c** Binding of purified LRP5 LDLa repeats fused with Fc to various fatty acids (non-linear fitting with Hill slope, $n = 3$ per group). Binding of these fatty acids to Fc was subtracted. AU, arbitrary unit. **d** Binding of purified LRP5 LDLa repeats fused with Fc to EPA or EPA ethyl ester (EPA-EE) ($n = 8$ per group). Data in this figure are all presented as mean ± sem with $p$ values (Student's $t$-test, Two-tailed, unpaired).

(Supplementary Fig. 3d). We found that CM containing LRP5 LDLa-Fc showed strong binding to EPA and DHA and to a lesser extent to ALA, ARA, and LA (Fig. 4a). Additionally, LRP5 LDLa-Fc showed no detectable specific binding for saturated and monounsaturated fatty acids, including oleic acid, palmitoleic acid, myristoleic acid, stearic acid, or myristic acid (Fig. 4b and Supplementary Fig. 3d). We also prepared CM containing Fc-fused individual repeats (R1, 2, and 3) and performed the EPA-binding assay. The R3 repeat showed detectable EPA binding (Supplementary Fig. 3e). Although R1 or R2 repeat alone showed no detectable binding to EPA, R1-2 showed EPA binding (Supplementary Fig. 3e). Thus, each of the three repeats may have certain affinities for EPA. Nevertheless, LDLa-Fc which contains all three repeats probably has the highest affinity for EPA because it showed the strongest binding (Supplementary Fig. 3e).

To further characterize the binding affinities of LDLa-FC to the fatty acids, dose-dependent binding of purified LRP5 LDLa-FC (Supplementary Fig. 3f) to EPA, DHA, ARA, LA, ALA and OL was performed (Fig. 4c). LDLa-Fc bound to EPA and DHA with the Kd values of ~69 and ~74 nM, respectively, and to ARA with a Kd value of ~479 nM (Fig. 4c). The affinities for ALA and LA were unable to be determined due to the failure to achieve saturation binding, whereas binding to OL was barely detectable (Fig. 4c). These results are consistent with the binding results using CM. The binding of LDLa-Fc to EPA was further validated by the isothermal titration calorimetry (ITC) assay, which revealed a Kd of 120 nM for the interaction (Supplementary Fig. 3g). We also purified Fc-fused LRP6 LDLa repeats (Supplementary Fig. 3f). LRP6 LDLa repeats showed no detectable binding to EPA (Supplementary Fig. 3h). Finally, we tested if LRP5 LDLa-Fc could bind to esterified PUFAs and found that it failed to bind to EPA ethyl ester (EPA-EE), an esterified form of EPA (Fig. 4d), suggesting that LRP5 may only transport non-esterified PUFAs.

## LRP5 transports EPA to intracellular compartments via internalization

Next, we examined the intracellular distribution of LRP5-transported EPA by staining neutrophils with an anti-PUFA antibody. This antibody showed strong binding to EPA and DHA, weak but detectable binding to ALA, ARA, and LA, and non-detectable binding to the tested monosaturated (oleic acid, palmitoleic acid and myristoleic acid) or unsaturated fatty acids (myristic acid and stearic acid) (Supplementary Fig. 4a). It also recognizes phosphatidylcholines containing PUFAs (EPA or DHA), but not phosphatidylcholine containing a saturated fatty acid (MA) (Supplementary Fig. 4b). These results together suggest that this PUFA antibody can detect non-esterified PUFAs (preferably EPA and DHA) as well as esterified PUFAs in phospholipids. To further evaluate the specificity of this antibody, we also tested a number of EPA metabolites, including prostaglandin E3, resolving E1, 18-HEPE, and Leukotriene B5. Our results showed that this antibody could not recognize any of these metabolites (Supplementary Fig. 4c). When this antibody was used to stain neutrophils isolated from mice on the essential fatty acid-free diet, it produced a stronger signal in cells cultured with EPA than those without (Fig. 5a Panels *a* vs. *b* and Supplementary Fig. 4d). PUFA antibody staining was partially colocalized with anti-TNG38 staining (a Golgi marker), suggesting that the antibody can be used to detect intracellular localization of loaded PUFAs in their unesterified and/or lipid forms. LRP5-deficiency markedly reduced anti-PUFA staining intensity, when normalized against TGN staining intensity, compared to that in WT cells (Fig. 5a, Panels *b* vs *c* and 5b), confirming that the anti-PUFA antibody detects LRP5-transported EPA.

To carry out a more unbiased analysis, we performed flowcytometric imaging analysis of a large number of cells and found that LRP5-deficient neutrophils from mice on the essential fatty acid-free diet and cultured with EPA showed significant reductions in anti-PUFA staining

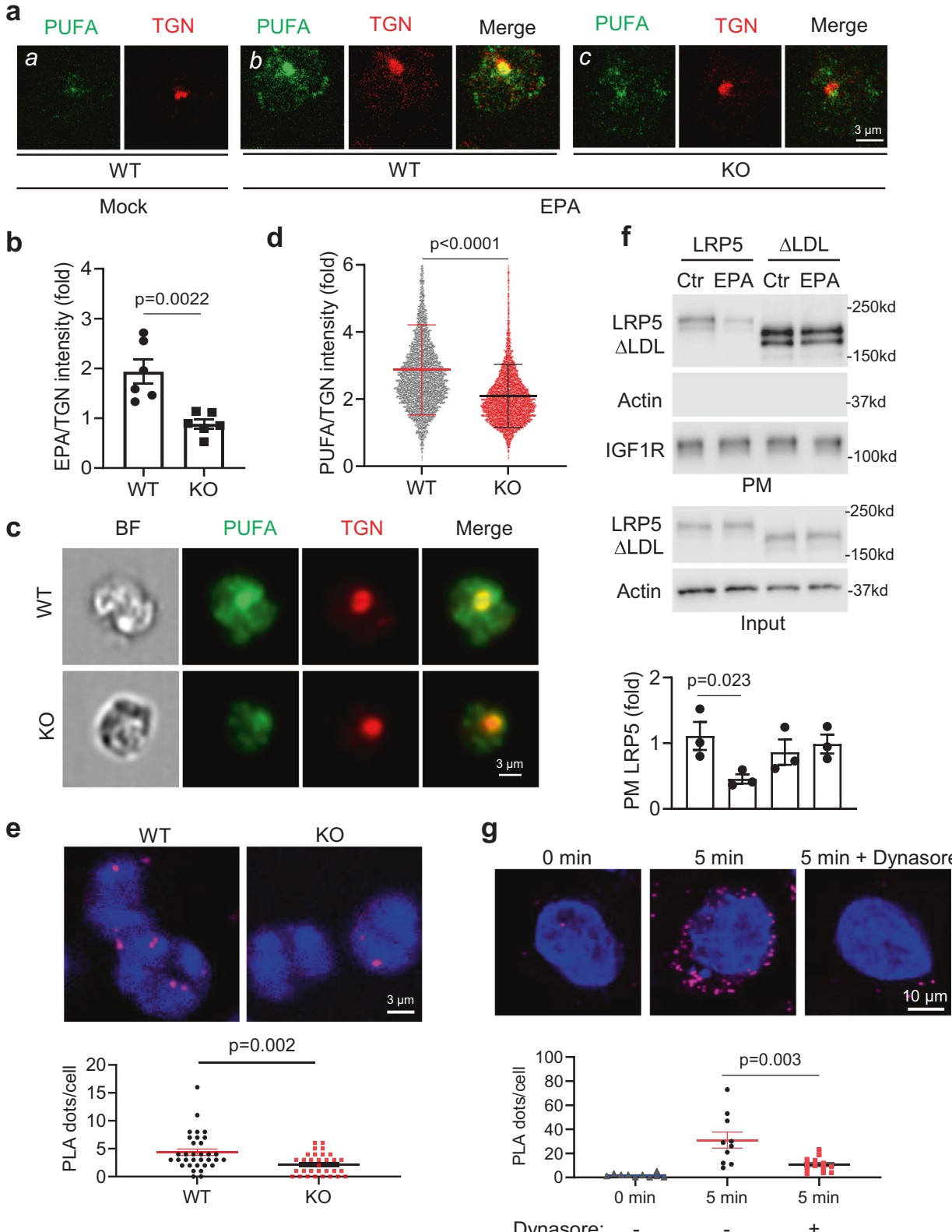

intensity normalized against anti-TGN38 staining intensity over that of the corresponding WT cells (Fig. 5c, d). By contrast, LRP5-deficiency had no significant effects on fluorescent intensity of Bodipy-palmitate normalized against anti-TGN38 staining in cells that were cultured with Bodipy-palmitate (Supplementary Fig. 4e). In addition to the localization at Golgi, LRP5-transported PUFAs were also detected at lysosomes, as LRP5-deficiency significantly reduced the co-

localization of the anti-PUFA and anti-LAMP1 (a lysosomal marker) antibodies (Fig. 5e).

The LDL receptor transports lipoproteins via internalization. We tested if EPA could promote LRP5 internalization using the LRP5 KO HEK293T cells re-expressing WT LRP5 or LRP5 lacking the LDLa repeats. We found that EPA reduced the amount of cell surface WT LRP5 without affecting the total LRP5 protein amounts (Fig. 5f).

**Fig. 5 | LRP5 transports PUFAs to intracellular compartments via internalization. a, b** Neutrophils from WT and neutrophil-specific LRP5 KO mice on the essential fatty acid-free diet and cultured with or without EPA (100 μM) for 3 h were stained with the anti-PUFA and anti-TGN38. Quantification of anti-PUFA staining intensity in Panels *b* and *c* with normalization against TGN staining intensity is shown in (**b**) (*n* = 6 cells per group from two independent experiments). **c, d** Flow cytometryt imaging analysis of neutrophils from WT and neutrophil-specific LRP5 KO mice on essential fatty acid-free diet and cultured with EPA, followed by staining with anti-PUFA and anti-TGN38. Quantification of anti-PUFA staining intensity after normalization against anti-TGN38 staining intensity is shown in (**d**) (*n* = 5969 cells for WT and *n* = 7014 cells for KO). **e** Neutrophils from WT and neutrophil-specific LRP5 KO mice on the essential fatty acid-free diet were cultured with EPA and subjected to proximity ligation assay (PLA) using antibodies for PUFA and LAMP1

(PLA fluorescence signal, magenta) and counterstained with DAPI (blue) (*n* = 32 cells for WT and *n* = 30 cells for KO from three independent experiments). **f** LRP5 KO HEK293T cells were transfected with the LRP5 or LRP5 ΔLDL expression plasmid and treated with or without EPA. Cell surface LRP5 proteins were detected by Western blotting after the precipitation of biotinylated cell surface proteins using avidin-columns (*n* = 3 per group from three independent experiments). PM plasma membrane. **g** LRP5 KO HEK293T cells with re-expressed WT LRP5 were cultured in the charcoal-filtered medium overnight, followed by incubation with EPA for the indicated period with or without dynasore (80 μM) treatment. The co-localization of EPA and LRP5 was detected using PLA (*n* = 10 cells for 0 min and 5 min without Dynasore, *n* = 13 cells for 5 min with Dynasore). Data in this figure are all presented as mean ± sem with *p* values (Student's *t*-test, Two-tailed, unpaired).

Consistent with the above observations, EPA did not reduce the amounts of surface LRP5 mutant lacking the LDLa repeats (Fig. 5f) or LRP6 (Supplementary Fig. 4f). These results suggest that EPA promotes internalization of LRP5, but not LRP6, and that this action depends on the presence of LRP5 LDLa repeats. The LDL receptor transport lipoproteins via clathrin-mediated endocytosis, which usually occurs within minutes[45,46]. We could detect intracellular co-localization of EPA and LRP5 at 5 min upon EPA addition to LRP5 knockout HEK293T cells re-expressing WT LRP5. In addition, treatment with dynasore, an inhibitor of clathrin-coated endocytosis, diminished this co-localization (Fig. 5g). Treatment with dynasore also led to a decrease in LRP5-mediated uptake of [14]C-EPA in HEK293T cells (Supplementary Fig. 4g). These results together suggest that LRP5 transports PUFAs via clathrin-mediated endocytosis.

### LRP5 is required for n-3 PUFA to suppress mTORC1

Next, we investigated mechanisms by which LRP5-transported PUFAs regulate neutrophils. n-3 PUFAs have been shown to regulate mTORC1[14–18], and our IPA pathway enrichment analysis of differentially expressed genes revealed by total RNA sequencing of isolated bone marrow LRP5-null and WT neutrophils showed mTORC1 related signaling pathways (EIF2, eIF4-p70S6K, and mTOR signaling pathways) among the altered ones (Supplementary Fig. 5a). Thus, we examined if LRP5 deficiency altered mTORC1 signaling activities. We found that phosphorylation of S6K and S6 was elevated in endogenously activated LRP5-null neutrophils isolated from inflammatory peritonea (Fig. 6a), as well as in naïve neutrophils isolated from bone marrow with or without chemoattractant stimulation (Fig. 6b), when compared to the corresponding WT control cells. There was a corresponding increase in the phosphorylation of 4EBP1, another mTORC1 downstream effector, in LRP5-deficient compared with WT neutrophils (Fig. 6a). By contrast, LRP6-deficiency did not affect S6K or S6 phosphorylation in neutrophils (Supplementary Fig. 5b), and LRP5 KO did not affect phosphorylation of AKT, ERK, or p38 (Fig. 6b). In neutrophils, mTORC1 signaling regulates NET formation[47,48]. Increased mTORC1 activity appeared to be responsible for increased NETs in the LRP5 KO neutrophils as LRP5-deficiency augmented histone citrullination (a NET surrogate marker) and this augmentation was abrogated by rapamycin (Supplementary Fig. 5c).

To determine if PUFAs underlie LRP5-deficiency-mediated activation of mTORC1 and NET formation in neutrophils, we examined the effect of PUFA-deprivation on mTORC1 signaling and histone citrullination. We found that PUFA-depleted WT neutrophils isolated from mice on the essential fatty acid-free diet had elevated S6K and S6 phosphorylation and histone citrullination compared to WT neutrophils isolated from mice on the normal diet (Fig. 6c, Lanes 1 vs 3). In contrast, PUFA-depletion had little effect on these events in LRP5 KO neutrophils (Fig. 6c, Lane 2 vs 4). Additionally, PUFA-depletion abrogated the differences between WT and LRP5-null neutrophils in these activities (Fig. 6c, Lanes 3 vs 4) that were observed between WT and LRP5-null neutrophils without PUFA-depletion (Fig. 6c, Lanes 1 vs 2).

These results suggest that PUFAs may suppress mTORC1 signaling in an LRP5-dependent manner. To further corroborate this conclusion, mice on the essential fatty acid-free diet were given back a PUFA via gavage feeding of EPA, which decreased S6K and S6 phosphorylation and histone citrullination in WT neutrophils (Fig. 6c, Lanes 3 vs 5) without altering these events in LRP5 KO neutrophils (Fig. 6c, Lanes 4 vs 6). Our LC-MS assay showed that gavage feeding of EPA to the mice on the essential fatty acid-free diet could bring the plasma EPA level to the normal range (Supplementary Fig. 2k). This LRP5-dependent inhibition of mTORC1 signaling by EPA was also observed in bone marrow neutrophils that were isolated from mice on the essential fatty acid-free diet and cultured with charcoal-filtered serum (deprived of fatty acids and lipids) (Supplementary Fig. 5d). Specifically, EPA inhibited S6K and S6 phosphorylation and histone citrullination in the WT, but not LRP5-null, neutrophils. In addition, DHA also showed significant inhibition of S6K and S6 phosphorylation and histone citrullination in the WT, but not LRP5-null, neutrophils (Supplementary Fig. 5d). ARA did not inhibit these mTORC1 signaling activities in neutrophils (Supplementary Fig. 5e), which is consistent with previous observations that ARA does not inhibit mTORC1 signaling in other cell types[14–18]. These results together indicate that n3-PUFAs inhibition of mTORC1 signaling and NET formation in neutrophils requires LRP5.

We also examined the effects of LRP5-transported PUFAs on mTORC1 signaling in HEK293T cells. LRP5-deficiency or PUFA withdrawal increased phosphorylation of S6K in HEK293T cells as well, and the addition of EPA only suppressed mTORC1 signaling in WT, but not LRP5 KO, HEK293T cells (Supplementary Fig. 6a). In addition, the re-expression of WT LRP5 in LRP5-KO HEK293T cells restored EPA-induced suppression of S6K phosphorylation (Supplementary Fig. 6b). FATP2 and CD36 also transport PUFAs[22,49]. Expression of CD36 resulted in a greater uptake of [14]C-EPA than that of LRP5, while FATP2 expression led to a similar uptake of [14]C-EPA to LRP5 expression (Supplementary Fig. 6c). However, the expression of either protein did not lead to significant EPA-induced suppression of S6K phosphorylation (Supplementary Fig. 6b). Moreover, the expression of LRP5, but not CD36 or FATP2, increased the co-localization of PUFAs with the lysosome marker LAMP1 (Supplementary Fig. 6d, h). These results together suggest that LRP5-transported PUFAs have different intracellular target preferences and functions from PUFAs transported by CD36 or FATPs.

### Unesterified EPA directly inhibits mTORC1 kinase activity

To investigate how LRP5-transported n-3 PUFAs suppress mTORC1 activity, we first investigated if PUFA derivatives downstream of COX2, lipoxygenase, or cytochrome P450 (CYP) enzymes may have a role in LRP5-mediated mTORC1 regulation. The elevation in mTORC1 signaling resulting from LRP5-deficiency was not blunted by the COX2 inhibitor celecoxib, the lipoxygenase inhibitor NDGA, or the CYPs inhibitor miconazole (Supplementary Fig. 6e–g), suggesting that COX2/lipoxygenase/CYPs-mediated PUFA metabolites are unlikely to be involved in LRP5-depednent mTORC1 regulation in

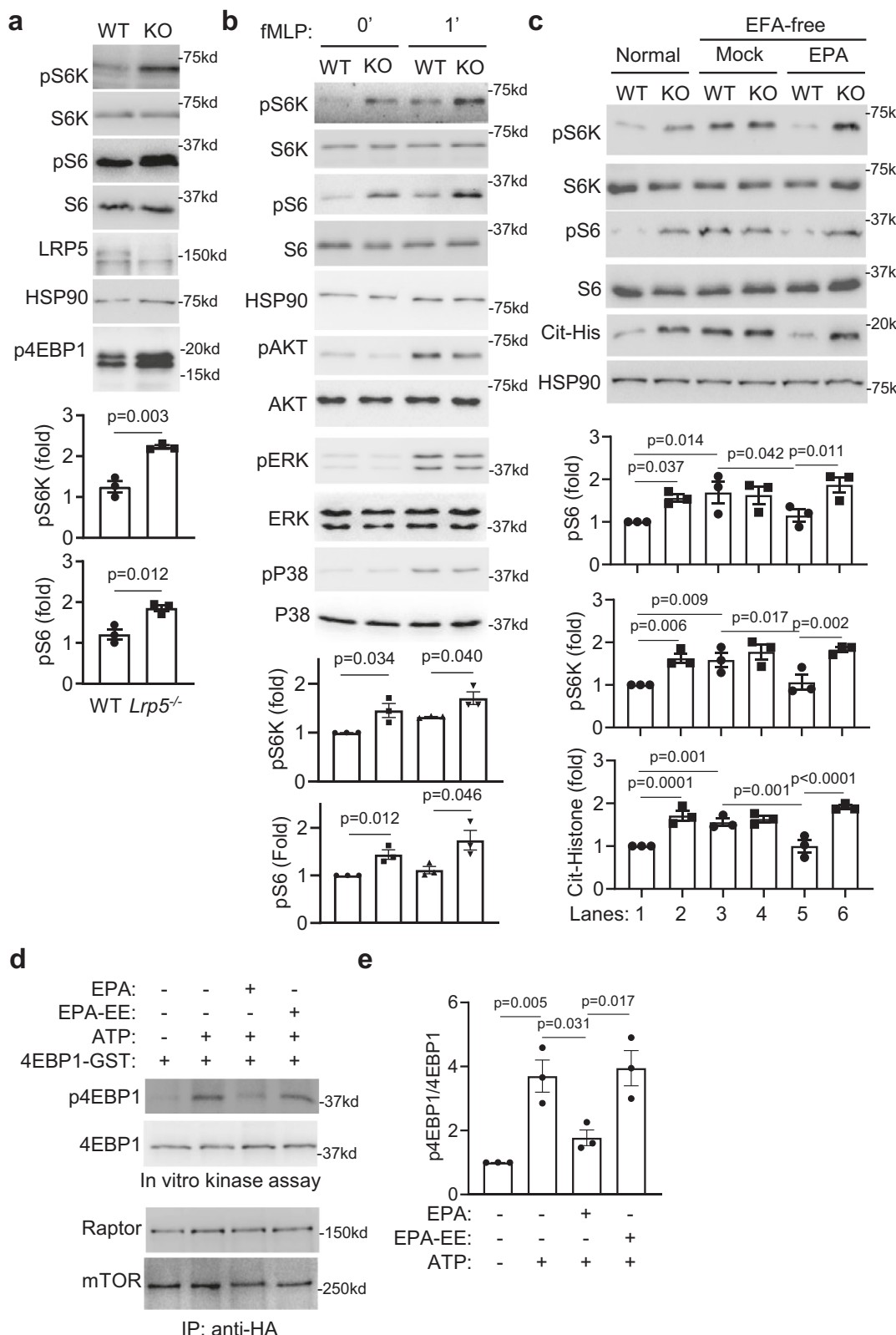

neutrophils. We next examined whether n3-PUFAs could directly act on mTORC1 and thus performed an in vitro kinase assay. The mTORC1 complex was pulled down from HEK293T cells expressing HA-tagged Raptor by an anti-HA antibody (Fig. 6d) and subjected to in vitro kinase assay using recombinant 4EBP1 fused with glutathione s-transferases (GST) as a substrate. The addition of EPA, but not EPA-EE, caused significant inhibition of 4EBP1 phosphorylation by

immunoprecipitated mTORC1 (Fig. 6d, e), suggesting that unesterified n3-PUFAs, but not esterified PUFAs, can directly inhibit mTORC1 kinase activity.

## Discussion

In this study, we identified LRP5 as being a selective transporter for unesterified PUFAs that plays an important role in PUFA accretion in a

**Fig. 6 | LRP5 is required for n-3 PUFA-mediated mTORC1 suppression in neutrophils.** Western analysis of WT and LRP5 KO neutrophils isolated from peritonea pretreated with thioglycolate for 12 h (**a**) or bone marrow neutrophils stimulated with fMLP (1 μM, (**b**)). Western qualification was normalized to corresponding total proteins (n = 3 per group from three independent experiments). **c** Bone marrow neutrophils isolated from WT and neutrophil-specific LRP5 KO mice fed on the normal diet or essential fatty acid-free diet with or without gavage of EPA were analyzed by Western blotting. Western blot qualification was normalized to HSP90

(n = 3 per group from three independent experiments). Data are presented as mean ± sem with p values (Student's t-test, Two-tailed, unpaired for (**a**, **b**); One-way ANOVA for (**c**)). **d** The mTORC1 complex was immunoprecipitated from HEK293T cells expressing HA-tagged Raptor via an anti-HA antibody and subjected to in vitro kinase assay with recombinant 4EBP1 as the substrate in the presence or absence of ATP, EPA, and EPA-EE. **e** The phosphorylation of 4EBP1 was quantified with normalization against total 4EBP1 (n = 3 per group from three independent experiments). Data are presented as mean ± sem with p values (One-way ANOVA).

number of different cells. Additionally, we showed that LRP5 was required for n-3 PUFA-mediated inhibition of mTORC1 in neutrophils, NK cells and HEK293T cells. We also demonstrated the functional importance of LRP5-transported PUFAs in neutrophils; LRP5-transported PUFAs protect mice from myocardial ischemia-reperfusion injury (Supplementary Fig. 7). This function of LRP5-transported PUFAs is consistent with epidemiologic studies indicating that PUFAs, particularly n-3 PUFAs, are beneficial for heart health[1,3–9].

While LRP5 had an important role for PUFA uptake in many different cells including neutrophils, NK cells and HEK293T, it is not the only PUFA transporter in these cells. LRP5-deficiency showed partial inhibition of $^{14}$C-EPA and $^{14}$C-ARA uptake in these cells to varying degrees (Fig. 3). This is understandable, given the existence of other fatty acid transport mechanisms that include passive diffusion and transport by CD36, FATPs and Mfsd2a. However, n-3 PUFA-mediated inhibition of mTORC1 signaling in HEK293T and neutrophils were almost completely dependent on LRP5 (Fig. 6c and Supplementary Fig. 6), indicating that LRP5-transported PUFAs have distinct functions from those transported by other mechanisms. Indeed, expression of CD36 or FATP2 increased EPA uptake, but failed to mediate inhibition of mTORC1 signaling by EPA (Supplementary Fig. 6b). Our result that unesterified, but not esterified, EPA could directly inhibit mTORC1 (Fig. 6d, e) provides a plausible mechanistic explanation for the distinct functions of these PUFA transport mechanisms. Because LRP5-mediated transport acts via LRP5 internalization (Fig. 5f), this mechanism can transport PUFAs to intracellular compartments including lysosomes in unesterified forms, where unesterified PUFAs in turn inhibit mTORC1. By contrast, the other mechanisms, including passive diffusion and transport by CD36 and FATPs, transport fatty acids across plasma membranes. These non-LRP5 transported PUFAs, even with the help of intracellular fatty acid-binding proteins (FABPs), may not be efficiently targeted to or largely esterified by the time they reach, some of intracellular compartments such as lysosomes where mTORC1 is regulated. This explains the observed effects of LRP5 KO on PUFA presence in Golgi and lysosomes (Fig. 5c–e) and the failure of CD36 or FATP2 expression to increase lysosomal localization of PUFAs and inhibit mTORC1 signaling (Supplementary Fig. 6b, d, h). Thus, this study demonstrates that different fatty acid transport mechanisms lead to different fates of transported fatty acids and distinct functional outcomes.

LRP5 is also distinguished from other known PUFA transporters for its selectivity to unesterified PUFAs. This molecular selectivity is likely due to the high affinity binding of LRP5 LDLa repeats for PUFAs compared with other fatty acids. LRP5 and LRP6 differ in the amino acid sequences of their respective repeats, which may also explain the inability of LRP6 to bind to and transport PUFAs.

Our study also provided examples for the functional importance of LRP5-transproted PUFAs. Consistent with the epidemiological evidence of n-3 PUFAs being anti-inflammatory, LRP5-mediated accretion of n-3 PUFAs in neutrophils suppresses NET formation by inhibiting mTORC1 signaling. NETs are known to mediate tissue injury during inflammation including myocardial ischemia-reperfusion injury[50–56]. Thus, the increased mTORC1 signaling and NET formation likely exacerbated myocardial ischemia-reperfusion injury in the neutrophil-specific LRP5 KO mice. Although our results also indicate that COX2/

lipoxygenase/CYPs-mediated PUFA metabolites are unlikely to be involved in LRP5-mediated mTORC1 regulation, they do not exclude the additional involvement of LRP5-transported PUFAs in other cellular effects. In addition, there could be mTORC1-independent actions of LRP5-transported PUFA in neutrophils, which would need to be further characterized in future studies. Knowing that LRP5-deficiency affects PUFA uptake in a broad range of cells, future studies also would need to investigate the importance of this LRP5 transport function in additional cell types including NK cells for its contribution to LRP5 deficiency-related phenotypes.

## Methods
### Mice
All animal experiments were performed with the approval of Institutional Animal Care and Use Committee (IACUC) at Yale University. The LoxP-floxed *Lrp5* (*Lrp5*$^{fl/fl}$) and *Lrp6* (*Lrp6*$^{fl/fl}$) mice were obtained from Bart Williams[57]. The *Lrp5*$^{fl/fl}$ and *Lrp6*$^{fl/fl}$ mice were back-crossed with C57BL/6 N mice for more than seven generations before being intercrossed with Lyz2-Cre (#004781), Rosa26-CreER2 (#008463), or Mrp8-Cre (#021614) mice (Jackson lab), followed by additional backcrossing with C57BL/6 N mice for more than ten generations. Ncr-1-Cre *Lrp5*$^{fl/fl}$ mice were described previously[34]. Wildtype C57BL/6N mice were purchased from Envigo (#044). The mice were housed under specific-pathogen-free conditions in Yale Animal Resources Center facilities and under 12 h light/dark cycles at 68–79 °F and 30–70% humidity. Mice in all experiments were age- and gender-matched. Sample sizes used in animal experiments were based on the empirical determination in preliminary experiments. For deprivation of PUFAs, mice were placed on an essential fatty acid-free Diet (TD.84224, Envigo) for 4 days. For PUFA refeeding, 100 mg/kgbw PUFAs were delivered to mice four times via oral gavage every 12 h just before experiments.

### Reagents and constructs
Antibodies to the following antigens were acquired commercially: Citrullinated-Histone-H3 (Abcam, ab5103), Myeloperoxidase/MPO (R&D, AF3667), LAMP-1 (Santa Cruz, sc-20011), TGN38 (Santa Cruz, Sc27680), PUFA (Cloud-Clone, PAO623Ge01), Beta-Catenin (BD, 610154), p-AKT473 (Cell Signaling Technology, 4060), mTOR (Cell Signaling, 2983), LAMP1 (Santa Cruz, sc-19992), PerCP-Cy5.5-Ly6G (BD, 560602), Pacific blue-CD11b (Biolegend 101223), Lrp5 (Cell Signaling Technology, 5731), Lrp6 (Cell Signaling Technology, 3395), Phospho-S6 (Cell Signaling Technology, 4858), total S6 (Cell Signaling Technology, 2217), Phospho-p70 S6K (Cell Signaling Technology, 9234), total p70 S6K (Cell Signaling Technology, 2708), Phospho-4E-BP1 (Cell Signaling Technology, 2855), total 4E-BP1 (Cell Signaling Technology, 9452), IGF-I Receptor β (Cell Signaling Technology, 9750), Beta-Actin (Proteintech, 66009), and HSP90 (Proteintech, 13171). All flow cytometry antibodies were used with 1:100 dilution. All western antibodies were used with 1:1000 dilution. The following reagents were also acquired commercially: fatty acids were all from Cayman Chemical Company, rapamycin (Millipore Sigma, 553210), formyl-Met-Leu-Phe (fMLP) (Sigma, F3506), Nordihydrogualaretic acid (Cayman Chemical, 70300), Celecoxib (Cayman Chemical, 10008672), GW9508 (Cayman Chemical, 10008907), Miconazole (Selleck Chemicals, S2536) and pFUSE-hIgG1-Fc2 (InvivoGen).

## Murine heart ischemia-reperfusion model

The mice were anesthetized with 5.0% isoflurane, intubated, and ventilated using a rodent ventilator. Anesthesia was maintained by inhalation of 1.5% to 2% isoflurane added to 100% oxygen. Body temperature was maintained at 37 °C. The hearts were exposed. Ischemia was achieved by ligating the left anterior descending coronary artery (LAD) using an 8–0 silk suture with a section of PE-10 tubing placed over the LAD. After occlusion for 45 min, reperfusion was initiated by releasing the ligature and removing the PE-10 tubing. The chest wall was closed, the animal extubated, and body temperature was maintained at 37 °C. Mice were re-anesthetized 24 h later. The heart was surgically exposed, and the LAD was reoccluded by tying the suture in order to define the ischemic area at risk for myocardial infarction and the non-ischemic area. The non-ischemic myocardium was perfused with 1% Evans blue that was infused into the aorta in a retrograde fashion. Hearts were excised, sliced into five 1-mm cross-sections with the aid of an acrylic matrix (ZIVIC Labs). The heart sections were incubated with 1% triphenyl tetrazolium chloride solution (TTC, Sigma-Aldrich) at 37 °C for 15 min. The non-ischemic area was stained blue, and within the ischemic region, residual viable myocardium was stained red and necrotic regions of myocardial infarction were unstained and appeared white. Using Image J software, the area of myocardial infarction was quantified and calculated as a percent of the myocardial area at risk, and the area at risk was calculated as a percent of the total LV area in each section.

## Neutrophil preparation

Murine neutrophils were isolated from bone marrows or peritoneal lavage. Briefly, bone marrow cells or peritoneal lavage cells (3% Thioglycolate induced overnight) collected from mice were treated with the ACK buffer (155 mM $NH_4Cl$, 10 mM $KHCO_3$ and 127 μM EDTA) for red blood cell lysis, followed by a discontinuous Percoll density gradient centrifugation. Neutrophils were collected from the band located between 81% and 62% of Percoll.

## Flow cytometry

Single-cell suspension was prepared, and the cell concentration was adjusted to $10^7$ cells/ml in staining buffer (1% BSA in PBS). Cells were pre-incubated with mouse Fc blocker (BD 553142, 1 μg/million cells in 100 μl) on ice for 10 min, followed by incubation with fluorescent-labeled primary antibodies on ice for 30 min in the dark. Cells were then washed with staining buffer twice and resuspended in staining buffer for flow cytometry analysis.

## Internalization assay

HEK293T cells were treated with mock or fatty acids (100 μM) in a culture medium for 3 h. The cells were washed once with 1% fatty-acid-free BSA in PBS, followed by twice washing with ice-cold PBS. Cell surface proteins were biotinylated with 0.5 mg/ml EZ-link-Sulfo-NHS-SS-Biotin (Thermo Fisher, A39258) in a PBS buffer with 2.5 mM $CaCl_2$ and 1 mM $MgCl_2$ on ice for 30 min. The reaction was stopped by the addition of PBS containing ice-cold 50 mM $NH_4Cl$, followed by repeated washes with ice-cold PBS. The cells were then lysed in a buffer containing 1.25% Triton X-100, 0.25% SDS, 50 mM Tris-HCl, pH 8.0, 150 mM NaCl, 5 mM EDTA, 5 mg/ml iodoacetamide, 10 μg/ml PMSF, and the Roche proteinase inhibitor cocktail. After centrifugation, aliquots were taken as lysate controls, and the rest of the supernatants were used in pulldowns with NeutrAvidin beads (Thermo Fisher, 29200), followed by western blotting analysis.

## RNAseq and analysis

CD11b and Ly6G double-positive neutrophils were sorted out from the mouse bone marrow cells. Total RNA was isolated from the neutrophils using RNeasy Plus Mini Prep (Qiagen, 74134). The quality of RNA samples was measured using the Agilent Bioanalyzer. Then, RNA-seq libraries were prepared with the TruSeq stranded total RNA library prep kit (Illumina). single-end sequencing (75 bp) was performed on an Illumina HiSeq 2500 instrument at Yale Center for Genome Analysis. RNA-seq reads were aligned to the mouse reference genome (MM9) and gene expression was quantified by running the DESeq2 pipeline (default parameters). Differentially expressed genes with adjusted $p$-value less than 0.05 from each comparison were used for signaling pathway analysis in IPA (Ingenuity Pathway Analysis). The raw data has been deposited at Gene Expression Omnibus (Accession number: GSE195678).

## $^{14}$C-fatty acid Uptake assay

$^{14}$C-labeled Eicosapentaenoic acid (0.2 μM, $^{14}$C-EPA, Moravek, MC2217), $^{14}$C-labeled arachidonic acid (0.2 μM, $^{14}$C-ARA, Moravek, MC364) or $^{14}$C-labeled oleic acid (0.2 μM, $^{14}$C-OL, American Radiolabeled Chemicals, ARC0297) was pre-mixed with or without 1% fatty-acid-free BSA (FAF-BSA) in RPMI1640 at 37 °C for 1 h. Cells were washed once with serum-free medium, followed by incubation with FAF-BSA conjugated $^{14}$C-fatty acids at 37 °C. After 1 h, cells were first washed twice with 1% FAF-BSA in PBS and then washed with PBS once. Finally, cells were resuspended in PBS, followed with the scintillation cocktail and counted by a liquid scintillation counter (PerkinElmer, Tri-CARB 2100TR).

## Flow imaging

EPA was pre-mixed with 1% fatty-acid-free BSA (FAF-BSA) in RPMI1640 at 37 °C for 1 h. Mouse primary neutrophils were incubated with 100 μM FAF-BSA conjugated EPA at 37 °C for 3 h, followed by twice washing with 1% FAF-BSA in PBS and once washing with PBS. Cells were fixed with 4% paraformaldehyde for 15 min at room temperature and then washed with PBS three times. Cells were then blocked with 1% FAF-BSA in PBS for 1 h at room temperature and then permeabilized with 0.03% saponin for 5 min. Primary antibodies were then diluted in TBST (1X Tris Buffered Saline, BioRad 1706435, containing 0.05% Tween) and applied to cells for overnight incubation at 4 °C. Cells were rinsed with PBS three times and incubated with diluted fluorochrome-conjugated secondary antibodies in TBST for 1 h at room temperature. Finally, cells were rinsed with PBS three times and resuspended in PBS with 2% FBS at a density of $5 × 10^7$ cells/ml. Samples were run on an Amnis ImageStream-X MarkII Imaging Flow Cytometer and analyzed by IDEAS 6.2.

## Immunostaining of heart sections

Hearts were perfused and fixed with 4% PFA (Santa Cruz, sc-281692) for 4–6 h on a shaker at 4 °C. They were then washed with PBS three times and perfused in 20% sucrose solution in PBS overnight at 4 °C. They were subsequently mounted in OCT embedding compound and frozen first at −20 °C and then at −80 °C. Tissue sections were prepared at 8-μm thickness with a cryostat and mounted onto gelatin-coated histological slides, which were stored at −80 °C. For immunostaining, slides were thawed to room temperature and fixed in pre-cold acetone for 10 min, then rehydrated in PBS for 10 min. The slides were incubated in a blocking buffer (1% horse serum and 0.02% Tween20 in PBS) for 1–2 h at room temperature, then incubated with primary antibodies, which were diluted in the blocking buffer, overnight at 4 °C. The slides were subsequently washed three times with PBS and incubated with secondary antibodies diluted in blocking buffer for 1 h at room temperature. After repeated washes, the slides were mounted with an anti-fade mounting media containing DAPI (Thermo Fisher, P36931) and visualized with a confocal microscope.

## Immunocytostaining

Primary neutrophils were then placed on poly-lysine-coated coverslips and fixed with 4% PFA for 10 min at room temperature followed by permeabilization with 0.03% saponin for 5 min at room temperature.

After being rinsed with PBS three times, cells were blocked with a blocking buffer (2% BSA in PBS) for 1 h at room temperature. Cells were then incubated with primary antibodies in blocking buffer at 4 °C for overnight. The next day, secondary antibodies with conjugated fluorescent probes (Alexa488 colored in green and Alexa633 colored in red in the figures) were 1:200 in blocking buffer and incubated with cells for 1 h at room temperature. Slides were prepared with the mounting medium containing DAPI and imaged under a confocal microscopy.

## Extracellular DNA measurement

Primary neutrophils were seeded into a poly-lysine coated 96-well plate at a density of $2 \times 10^5$ cells/well and treated with 10 ng/ml GM-CSF for 4 h. 1000 mU/ml micrococcal nuclease (NEB, M0247S) was applied to cells in a volume of 100 µl per well and incubated at 37 °C for 10 min. The reaction was stopped with 5 mM EDTA and the supernatant was collected and centrifuged at 200 $g$ for 8 min. The DNA in the supernatant was quantified using the Quant-iT Picogreen assay (Thermo Fisher, P7589).

## ELISA for anti-PUFA antibody and PUFA binding by LRP5 fragments

For detection of fatty acid binding by the anti-PUFA antibody, a 96-well white plate (NUNC polysorp, 437702) was coated with 1% fatty-acid-free BSA in PBS overnight at room temperature. The plate was rinsed once with PBS, followed by incubation with 100 µl of 0.5 µg/µl fatty acid in PBS at 37 °C for 1 h. The plate was then washed three times with PBS and incubated with the anti-PUFA antibody (1:1500 dilution in TBST). After 2 h later, the plate was rinsed with TBST for three times, followed by incubation with an HRP-conjugated goat anti-rabbit secondary antibody (Jackson ImmunoResearch, 111-035-144; 1:1000 dilution in TBST) for 30 min at room temperature. After three times washing with TBST, the plate was loaded with the SuperSignal pico plus chemiluminescent substrate (Thermo Fisher, 34580) and read by a plate reader (Perkin Elmer).

For detection of LRP5 fragment binding to PUFAs, a 96-well white plate was coated with 1% fatty-acid-free BSA in PBS overnight at room temperature. The plate was rinsed once with PBS, followed by incubation with a fatty acid at 37 °C for 1 h. The plate was then washed three times with PBS. The conditioned media or purified proteins were added to the plate. After 2 h of incubation at room temperature, the plate was rinsed with TBST for 3 times, followed by incubation with an HRP-conjugated goat anti-human secondary antibody (Jackson ImmunoResearch, 109-035-003; 1:1000 dilution in TBST) for 30 min at room temperature. After three times washing with TBST, the plate was loaded with the SuperSignal pico plus chemiluminescent substrate (Thermo Fisher, 34580) and read with a plate reader (Perkin Elmer).

## Shotgun lipidomic analysis

For the analysis, quadruplicate samples were prepared from the *LRP5*$^N$ KO and WT control mice. Each sample consists of 10 million peritoneal neutrophils pooled from two mice. The cells were washed in PBS containing 1% fatty acid-free BSA for three times and snap-frozen in liquid nitrogen before being stored at −80 °C. The frozen samples were shipped on dry ice to Lipotype GmbH (Dresden, Germany), where they were extracted and analyzed using the Shotgun Lipidomic Platform. For lipid extraction, Lipids were extracted using MS grade chloroform and methanol[58]. Samples were spiked with lipid class-specific internal standards prior to extraction. After drying and re-suspending in MS acquisition mixture, lipid extracts were subjected to mass spectrometric analysis. For spectra acquisition, Mass spectra were acquired on a hybrid quadrupole/Orbitrap mass spectrometer equipped with an automated nano-flow electrospray ion source in both positive and negative ion mode. For data processing and normalization, Lipid identification using LipotypeXplorer was performed on unprocessed (*.raw format) mass spectra[59]. For MS-only mode, lipid identification

was based on the molecular masses of the intact molecules. MSMS mode included the collision-induced fragmentation of lipid molecules and lipid identification was based on both the intact masses and the masses of the fragments. Prior to normalization and further statistical analysis lipid identifications were filtered according to mass accuracy, occupation threshold, noise and background. Lists of identified lipids and their intensities were stored in a database optimized for the particular structure inherent to lipidomic datasets. Intensity of lipid class-specific internal standards was used for lipid quantification. For data quality control, the dynamic range for cell culture samples was determined prior to analysis[58]. All the extractions and sample processing must be done without any incidents. The ionization quality must be satisfactory and no issues were encountered. The internal standard signal-to-noise ratio was in range of hundreds and more, indicating high quality of spectra. Based on these data, limits of quantification and coefficients of variation for the different lipid classes were determined. Limits of quantification are in the lower µM to sub-µM range, depending on the lipid class. The average coefficient of variation for a complete set of quantified lipid classes is around 10–15%. Each analysis is accompanied by a set of blank samples to control for a background and a set of quality control reference samples to control for intra-run reproducibility and sample specific issues. The technical reproducibility, as assessed by quality control reference samples (mammalian full blood) included in the same analytical run was very good, and blank samples included in each analytical batch (internal background control) did not contain any unusual background. In terms of Lipid nomenclature, depending on the nature of the raw data, lipid molecules may be identified as species or subspecies. Fragmentation of the lipid molecules in MSMS mode delivers subspecies information, i.e., the exact acyl chain (e.g., fatty acid) composition of the lipid molecule. MS only mode, acquiring data without fragmentation, cannot deliver this information and provides species information only. In that case, the sum of the carbon atoms and double bonds in the hydrocarbon moieties is provided. Lipid species are annotated according to their molecular composition as: Head group <sum of the carbon atoms in the hydrocarbon moiety>:<sum of the double bonds in the hydrocarbon moiety >For example PI 34:1 denotes phosphatidylinositol with a total length of its fatty acids equal to 34 carbon atoms, total number of double bonds in its fatty acids equal to 1. Lipid subspecies annotation contains additional information on the exact identity of their acyl moieties and their sn-position (if available). For example PI 18:1_ 16:0 denotes phosphatidylinositol with octadecenoic (18:1) and hexadecanoic (16:0) fatty acids, for which the exact position (sn-1 or sn-2) in relation to the glycerol backbone cannot be discriminated (underline "_" separating the acyl chains). On contrary, PC O-18:1/16:0 denotes an ether- phosphatidylcholine, where an alkyl chain with 18 carbon atoms and 1 double bond (O-18:1) is ether-bound to sn-1 position of the glycerol and a hexadecanoic acid (16:0) is connect via an ester bond to the sn-2 position of the glycerol (slash "/" separating the chains signifies that the sn-position on the glycerol can be resolved).

## Targeted LC-MS/MS analysis of phospholipids

Cells (3-10 million) were collected and washed in PBS containing 1% fatty acid-free BSA for three times. Cell pellet was re-suspended in 600 µl of ice-cold chloroform/methanol (2:1, v/v) containing 1 mM butylated hydroxytoluene (BHT) and incubated on ice for 30 min with occasional vortex mixing. Water (150 µl) was then added for 10 min. Samples were centrifuged at 2000 rpm for 10 min at 4 °C, and the organic layer (bottom layer) was collected to a new glass tube, whereas the aqueous phase (top layer) was re-extracted with 200 µl ice-cold chloroform/methanol (2:1, v/v). Organic phases (400 µl from first-round extractions and 150 µl from second-round extractions) were combined and dried by a vacuum evaporator. Samples were reconstituted with 150ul methanol:$H_2O$ (2:1, v/v) containing 1% formic acid.

Phospholipids were separated using an Agilent 1200 HPLC system before introduction into an API 3000™ Triple Quadrupole Mass Spectrometer (Applied Biosystems). PC was separated by an Inertsil HILIC column and was acquired in the positive ESI mode, while PE and PA were separated by an XTerra MS C18 Column (125 Å, 3.5 μm, 4.6 mm × 150 mm) and measured in the negative ESI mode. Collision energies ranged from 35 to 60 volts for various lipid species in MS/MS scan modes. Multiple reaction monitoring transitions of phospholipids were as follows: PC(O-36:5), 766.4 > 184.0; PC(36:5), 780.5 > 184.0; PC(38:5), 808.5 > 184.0; PC(O-38:5), 794.5 > 184.0; PA(18:0_20:5), 721.3 > 301.3; PA(20:1_20:5), 747.5 > 301.3; PE(P-18:0_20:5), 748.5 > 301.2; and PE(P-20:0_20:5), 776.5 > 301.2. Individual lipid species were quantified after normalization with spiked corresponding internal standards PC(14:0/14:0), PE(16:0-d9/16:0), and PA(16:0/18:1) and were shown as pmol per one million cells.

### Targeted LC-MS/MS analysis of deuterated EPA

Neutrophils were collected and washed in PBS containing 1% fatty acid-free BSA for three times. Cells were re-suspended in 100 μl of PBS and then added with 100ul of 2.5 N KOH/Meoh (1:4, v/v). Cell lysates were then incubated at 72 °C for 15 min and acidified by adding 25 μl formic acid followed by adding 225 μl chloroform. The bottom chloroform layer (150 μl) was transferred to a glass vial and was dried. The dried extracts were reconstituted with 100 μl chloroform:methanol (1:4, v/v) and separated by an a XTerra MS C18 Column (125 Å, 3.5 μm, 4.6 mm × 150 mm) with mobile phases of 20 mM Ammonium acetate (pH = 9) and Methanol. Deuterium-labeled EPA was acquired in the negative ESI mode. Collision energies was 40 volts in MS/MS scan modes. Multiple reaction monitoring transitions for EPA-d5 was 306.2 > 262.2.

### Purification of LRP5 LDLa-Fc and Fc proteins

The Expi293 Expression System from ThermoFisher Scientific was used to express the Fc fusion proteins according to manufacturer's instructions. The Expi293 cells were transfected using the ExpiFectamine 293 Transfection Kit. Protein expression in culture supernatant was confirmed 72 h post-transfection and harvested 7 days post-transfection. Cell debris and macrovesicles were removed from conditioned media by centrifugation at $380 \times g$ for 20 min at 4 °C. Recombinant human IgG1-tagged proteins were then purified by the Protein A/G agarose (Santa Cruz Biotechnology). Proteins were eluted with 100 mM glycine pH 3.0 and collected in tubes containing 1 M Tris buffer, pH 8.5. After dialysis with PBS overnight at 4 °C, protein concentrations were determined by the Bradford assay.

### NK cell purification and culture

Mouse primary NK cells were isolated from the spleens of mice (8 weeks old) by using the NK cell isolation kits according to the manufacturer's instructions (Miltenyi Biotec #130-090-864). Primary NK cells were cultured in RPMI-1640 (Gibco, 11875-093) supplemented with 10% FBS, penicillin (100 U/ml), streptomycin (100 μg/ml), 2-mercaptoethanol (500 μM) and HEPES (10 mM) at 37 °C supplemented with 5% $CO_2$ in the presence of recombinant murine IL-15 (50 ng/ml).

### Proximity ligation assay

Primary neutrophils were placed on poly-lysine-coated coverslips and fixed with 4% PFA for 10 min at room temperature followed by permeabilization with 0.03% saponin for 5 min at room temperature. Cells were then incubated with pairs of antibodies (one from mouse and one from rabbit). PLA was carried out with the Duolink reagents from Sigma-Aldrich according to the manufacturer's directions. Briefly, cells were incubated with a pair of suitable PLA antibody probes in a humidified chamber, which were then subjected to ligation and amplification with fluorescent substrate at 37 °C. The slides were mounted in mounting solution with DAPI. Cells were imaged with a Leica SP5 confocal microscope. Images were processed and quantified by Image J software as described[60].

### Isothermal titration calorimetry

Isothermal titration calorimetry was performed using TA instruments Nano-ITC machine at Yale core facility. 200 μM EPA was injected into 20 μM purified proteins. Data was analyzed using NanoAnalyze software from TA instruments.

### Statistical analysis and study design

Minimal group sizes for mouse studies were determined by using power calculations with the DSS Researcher's Toolkit with an α of 0.05 and power of 0.8. Animals were grouped unblinded, but randomized, and investigators were blinded for most of the qualification experiments. No samples or animals were excluded from analyses. Assumptions concerning the data including normal distribution and similar variation between experimental groups were examined for appropriateness before statistical tests were conducted. Comparisons between two groups were performed by unpaired, two-tailed t-test. Comparisons between more than two groups were performed by one-way ANOVA, whereas comparisons with two or more independent variable factors by two-way ANOVA using Prism 9.0 software (GraphPad). Statistical analyses were based on biological replications. $P < 0.05$ is considered as being statistically significant. All of the experiments were repeated at least twice, and representative ones are shown.

### In vitro mTORC1 kinase assay

Immunoprecipitation of mTOR complex 1 and kinase assays were performed as described previously[61]. Briefly, HEK293T cells transfected with HA-tagged Raptor were cultured in DMEM medium with charcoal-stripped serum for 16hrs. Cells were then lysed on ice for 20 min in the lysis buffer (40 mM HEPES [pH7.4], 120 mM NaCl, 2 mM EDTA, 0.3% CHAPS) supplemented with protease inhibitors (Complete Mini, Roche) and phosphatase inhibitor (PhosSTOP, Roche). After the centrifuge, the supernatant was incubated with the HA antibody at 4 °C for 90 min, followed by incubation with protein A/G-plus-agarose (Santa Cruz) for another hour. Immunocomplex was washed three times in the lysis buffer and twice with mTORC1 kinase buffer (25 mM HEPES [pH7.4], 50 mM KCl, 10 mM $MgCl_2$). For the kinase assay reaction, immunocomplexes were incubated in a volume of 30 μl for 30 min at 37 °C in the mTORC1 kinase buffer containing 200 ng of bacterially purified GST-4EBP1 (MyBioSource), 500 μM ATP with 100 μM EPA, EPA-EE, or corresponding control buffer. The reaction was stopped by adding 10 μl of 4x SDS sample buffer, followed by Western analysis.

### Reporting summary

Further information on research design is available in the Nature Portfolio Reporting Summary linked to this article.

## Data availability

The RNAseq data has been deposited to Gene Expression Omnibus and is publicly available. The accession number is GSE195678. Source data are provided with this paper.

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

## Acknowledgements

We thank YCGA for performing the next generation sequencing. We also thank the National Institutes of Health for funding support. This work is supported by NIH grants to D.W. (RO1CA274735 & R35HL135805).

## Author contributions

W.T., Y.L., Q.Y., A.L., S.C., S.M. performed experiments; W.T., Y.L., D.W. designed experiments and performed data analysis; L.Y. provided scientific advice; W.T., Y.L., D.W. prepared the manuscript; All authors discussed the results and commented on the manuscript.

## Competing interests

The authors declare no competing interests.
