## [Peer Review File · Nature Communications]

LDL Receptor-Related Protein 5 Selectively Transports Unesterified Polyunsaturated Fatty Acids to Intracellular CompartmentsREVIEWER COMMENTS

Reviewer #1 (Remarks to the Author):

In the current manuscript, Tang et al investigate the role of LRP5 in PUFA transport to intracellular compartments. The authors use a combination of in vivo and in vitro approaches to demonstrate LRP5 transports unesterified PUFA to translocate to specific intracellular compartments and inhibit mTORC1. The overall idea is interesting and novel. There are some major concerns that need to be addressed in the current manuscript.

Concerns:

- 1) The authors assess intracellular translocation of PUFA using an antibody based approach. This is a major weaknesses in the study. The information regarding this antibody is limited (in both the paper and scientific literature) and the results need to be validated. It is unclear what the authors are detecting or what the antibody is binding to in the experiments.
- 2) The study only used a limited approach to assess the impact of PUFA metabolites, ie., COX-2 inhibitor and LOX inhibitor. Further rationale, and and experiments are critical, such as inhibitors, including those targeting CYP system. These are warranted as there is a growing body of evidence suggesting it is the metabolites of n-3 and n-6 PUFA that are mediating much of the responses attributed to PUFAs.
- 3) The rationale and validation of the in vivo diet is needed. The relatively short duration and route of delivery was questioned. How do the authors know there was a significant reduction or increase in EFA levels in the animals, did they change in the heart itself or attenuate the immune response?
- 4) The cell studies utilize single concentrations of PUFA (100uM), which is considered high. A concentration response and rationale would be beneficial. As well, the single time point (3h) was utilized with the exception of FigS6D that incubated cells for only 5min to make the conclusion CD36 and FATP2 were having a minor role. Further experiments/rationale are needed.
- 5) It is important include the correct loading controls for all immunoblots, specifically the 'total' proteins for the respective 'phosphorylated' proteins should be done (ie., p~Akt to total Akt, pS6 to total S6, etc) and rationalize why normalize to HSP90?

Reviewer #2 (Remarks to the Author):

This manuscript by Tang et al identified LRP5 as PUFA transporter in cells, predominantly neutrophils. The manuscript is interesting and contain a number of insightful experiments to show this. A functional

aspect is also included, showing that when neutrophils lack LRP5 and cannot take up PUFAs, neutrophils are more activated, and this causes a worse outcome in a model of cardiac IR. However, the structure of the manuscript doesn't allow it to flow and show the data/story in the best way possible. Thus, a restructure and additional experiments are required.

Major comments:

1. The functional data in Figure 1 and start of figure 2 should be moved to the end of the manuscript and expanded on. Moving Figure 1 to the end, the authors should include images (Fig S1 E) and a quantification of this (NETosis/MPO) along with the histology (Fig S1 A) in the main figure. The data in figure 2A should also show these same readouts on NETosis and MPO. A full lipidomic analysis should be included in the neutrophils from the KO and essential FA free (EFA) diet fed mice, not just select lipids in the KO mice such as included in Fig S2 D.
2. The manuscript should begin with the lipidomic data in figure 2B onwards, with a more complete/ in-depth analysis. To start with perhaps volcano plots would be better to show the overall lipid species that are changing between the WT and KO neutrophils.
3. Figure 3: It would be important to show the expression levels of LRP5 across all these cell types by western blot. The conclusion to the end of the figure in the results section should be removed or reworded as it shows myeloid/innate cells seem to use LRP5 to take up PUFAS, but not lymphoid cells (i.e. T cell). The authors should also include B cells in this analysis.
4. Figure 6: The data presented here is very interesting. How were the neutrophil lipidomes altered in the WT and KO mice between resting vs activated (i.e. in A; WT vs KO in the inflammatory model and/or B; control vs fMLP).

Reviewer #3 (Remarks to the Author):

The authors did a great deal of work with innovative findings presented. I have some concerns regarding lipidomics analyses and I have the following comments:

1. The authors described in the lipidomics method section that both shotgun lipidomics and targeted LC-MS/MS lipidomics were performed. With MRM transitions described for only 8 phospholipids, I assumed that these 8 phospholipids were measured by targeted LC-MS/MS and the rest of the features which were listed in the Supplementary Table 1 were measured by shotgun lipidomics. Please clarify which lipid targets were measured by which platform since it is very confusing. In addition, please provide analytical information about sample extraction for shotgun lipidomics or refer to a method paper if available.
2. For targeted LC-MS/MS analysis, the cell count ranged from 3-10 million which varied among different biological samples, how was sample-based normalization performed?
3. The authors should indicate how quality control was performed for lipidomic analyses including batch design, the use of a pooled sample for quality control and data pre-processing strategies. Were biological samples randomized for LC-MS analysis? Which correction strategy was performed for isotopic interferences?
4. In the paragraphs describing Targeted LC-MS/MS analysis of phospholipids and Targeted LC-MS/MS analysis of deuterated EPA, please clarify the exact volume of organic layer transferred and add column

size. If available, please refer to a lipidomics method paper.

5. Please carefully check lipid nomenclature in supplementary Table 1. Consider using Lipid Maps nomenclature <https://www.lipidmaps.org/> It is not clear that what 0 following the semicolon means.

Define the data unit (or unit-free) for each feature.

POINT-BY-POINT RESPONSES TO REVIEWERS' COMMENTS:

We greatly appreciate the overall positive feedback on the novelty and significance of our work from all three reviewers. We are also very grateful for the insightful and constructive comments and suggestions provided by the reviewers, which helped us to further improve our manuscript.

Responses to the comments of reviewer #1

Overall Comment:

In the current manuscript, Tang et al investigate the role of LRP5 in PUFA transport to intracellular compartments. The authors use a combination of in vivo and in vitro approaches to demonstrate LRP5 transports unesterified PUFA to translocate to specific intracellular compartments and inhibit mTORC1. The overall idea is interesting and novel. There are some major concerns that need to be addressed in the current manuscript.

Response: We thank the reviewer for recognizing the scope of our study interesting and novel. We also appreciate the constructive suggestions provided by the reviewer.

Concern 1:

The authors assess intracellular translocation of PUFA using an antibody based approach. This is a major weaknesses in the study. The information regarding this antibody is limited (in both the paper and scientific literature) and the results need to be validated. It is unclear what the authors are detecting or what the antibody is binding to in the experiments.

Response: We agree with the reviewer that there is limited published information on this anti-PUFA antibody. We had the same concern and had done an array of validation work before the antibody was used in tightly controlled experiments (i.e., to compare the LRP5 KO and WT cells that were deprived of PUFAs and subsequently added back with EPA).

In our original submission, we had tested the antibody for its specificity to two PUFAs (EPA and DHA), a monounsaturated fatty acid (oleic acid), and two saturated fatty acids (palmitic acid and stearic acid). Heeding the reviewer's recommendation for further validation of the antibody, we performed more comprehensive validation by testing additional fatty acids including additional PUFAs, monounsaturated and saturated fatty acids, which is shown in **Fig. S4A**. The antibody showed strong binding to EPA and DHA, weak but detectable binding to ALA, ARA, and LA, and non-detectable binding to the tested monosaturated or unsaturated fatty acids. Additionally, the antibody can also bind to phosphatidylcholines containing PUFAs (EPA or DHA), but not to the phosphatidylcholine containing a saturated fatty acid (MA) (**Fig. S4B**). These results together suggest that this PUFA antibody could detect non-esterified PUFAs (preferably EPA and DHA) as well as PUFAs in phospholipids. To further evaluate the specificity of this antibody, we also tested a number of EPA metabolites, including prostaglandin E3, resolving E1, 18-HEPE, and Leukotriene B5. Our results showed that this antibody could not recognize these metabolites (**Fig. S4C**).

Importantly, as mentioned earlier we only used the antibody under tightly controlled conditions, i.e., only being used to compare the subcellular localization of EPA between LRP5

KO and WT cells that were deprived of PUFAs and added back with EPA. Our data shown in **Fig 5A** (a vs b) and **Fig. S4D** demonstrated that cells cultured with EPA showed notably enhanced staining signals compared with cells cultured without EPA (the culture medium used here contains charcoal-filtered serum so fatty acids were deprived). The signals detected by this antibody here should primarily be from EPA or EPA-incorporated lipids (also possibly other EPA derivatives that this antibody may recognize).

Concern 2:

The study only used a limited approach to assess the impact of PUFA metabolites, ie., COX-2 inhibitor and LOX inhibitor. Further rationale, and and experiments are critical, such as inhibitors, including those targeting CYP system. These are warranted as there is a growing body of evidence suggesting it is the metabolites of n-3 and n-6 PUFA that are mediating much of the responses attributed to PUFAs.

Response: This is a very good suggestion that CYP system is one of the important pathways to metabolize PUFAs. CYPs metabolize PUFAs to either Ep-PUFAs (epoxygenase activity) or hydroxy-PUFAs (hydroxylase activity), which have important physiological roles, especially in the neuron system [1]. In the revised manuscript, we used Miconazole (MIC) to block CYP activity. MIC is a non-selective CYP inhibitor, which can block both CYP hydroxylase and epoxygenase [2]. Our result showed that MIC didn't affect mTORC1 signaling elevation resulting from LRP5-deficiency (**Fig. S6G**), which suggested that CYP-mediated PUFAs metabolites are unlikely to be involved in LRP5-dependent mTORC1 regulation in neutrophils.

Concern 3:

The rationale and validation of the in vivo diet is needed. The relatively short duration and route of delivery was questioned. How do the authors know there was a significant reduction or increase in EFA levels in the animals, did they change in the heart itself or attenuate the immune response?

Response: This is an insightful question. We conducted the analysis of the EPA levels in mice subjected to the normal diet, essential fatty acid-free (EFA-free) diet, and EPA refeeding after EFA-free diet using targeted LC-MS analysis. The results revealed a significant reduction in EPA levels in the plasma of mice on the EFA-free diet compared to those on the normal diet and showed refeeding with EPA could bring plasma EPA level to the normal range. This finding is now incorporated into the revised manuscript (**Fig. S2K**). Additionally, in response to Reviewer #2, we also performed the lipidomics analysis of neutrophils obtained from mice under EFA-free dietary conditions. In comparison to the lipidomics analysis of neutrophils from normal diet mice, we observed significant reductions in the contents of lipids containing PUFAs in cells from mice on the EFA-free diet. In contrast, the contents of lipids containing saturated and

monounsaturated acids remained largely unaltered between mice fed the normal diet and those on the EFA-free diet (**Fig. S2I&S2J**).

Our results revealed increased myocardial ischemia-reperfusion injury in neutrophil-specific (Mrp8-Cre driven) LRP5-KO mice compared to control mice, suggesting that LRP5 in neutrophils is responsible for the observed phenotypes. Furthermore, feeding with EFA-free diet not only elevated myocardial ischemia-reperfusion injury in control mice but also abrogated the differences between control and neutrophil-specific LRP5-KO mice (**Fig. 2A, S2A&S2B**), indicating the effects of PUFAs predominantly acted on neutrophils through LRP5. We also showed that EPA could reduce neutrophil mTORC1 signaling and NETs formation through LRP5 (**Fig. 6C & S6A**). Thus, we think neutrophil response to the PUFA levels plays an important role in the heart phenotypes that we observed in this study.

Concern 4:

The cell studies utilize single concentrations of PUFA (100uM), which is considered high. A concentration response and rationale would be beneficial. As well, the single time point (3h) was utilized with the exception of FigS6D that incubated cells for only 5min to make the conclusion CD36 and FATP2 were having a minor role. Further experiments/rationale are needed.

Response: It was reported that the free fatty acid level in mouse plasma is approximately 0.1 nmol/ul, equivalent to 100 uM [3]. And this concentration of 100 uM PUFAs is commonly employed in the scientific literature for in vitro cell stimulation [4, 5]. We incubated cells with EPA for 5min to detect EPA internalization (**Fig. 5G**), which occurs within minutes. For measuring downstream signaling (mTORC1), we usually incubated cells with EPA for 2-3 hrs [4, 5]. In our pilot experiments, we tested shorter treatments and lower doses of EPA, the condition used (100 μ M and 3 hours) appeared to give the strongest responses on mTORC1 signaling.

The 5min incubation for FigS6D in our previous figure legends was a typo, which has been corrected in the revised version. Cells were incubated with EPA for 2-3 hrs to determine the effects of CD36 and FATP2 on transporting PUFAs to the lysosome (**Fig. S6D**) and their effects on S6K phosphorylation (**Fig. S6B**).

Concern 5:

It is important include the correct loading controls for all immunoblots, specifically the 'total' proteins for the respective 'phosphorylated' proteins should be done (ie., p~Akt to total Akt, pS6 to total S6, etc) and rationalize why normalize to HSP90?

Response: We appreciate this suggestion. Our results in **Fig. 6A & 6B** demonstrated the protein contents of S6K and S6 were similar to those of the housekeeping protein HSP90 under our experimental conditions. Since HSP90 could be blotted with the phosphorylated proteins on the same membrane, we thought it would be more accurate to normalize phosphorylated proteins to HSP90. In our revised manuscript, we added the total protein immunoblots.

Responses to the comments of reviewer #2

Overall Comment:

This manuscript by Tang et al identified LRP5 as PUFA transporter in cells, predominantly neutrophils. The manuscript is interesting and contain a number of insightful experiments to show this. A functional aspect is also included, showing that when neutrophils lack LRP5 and cannot take up PUFAs, neutrophils are more activated, and this causes a worse outcome in a model of cardiac IR. However, the structure of the manuscript doesn't allow it to flow and show the data/story in the best way possible. Thus, a restructure and additional experiments are required.

Response: We thank the reviewer for considering our study interesting and insightful.

Concern 1:

The functional data in Figure 1 and start of figure 2 should be moved to the end of the manuscript and expanded on. Moving Figure 1 to the end, the authors should include images (Fig S1 E) and a quantification of this (NETosis/MPO) along the with the histology (Fig S1 A) in the main figure. The data in figure 2A should also show these same readouts on NETosis and MPO. A full lipidomic analysis should be included in the neutrophils from the KO and essential FA free (EFA) diet fed mice, not just select lipids in the KO mice such as included in Fig S2 D.

Response: We greatly appreciate the reviewer's suggestions. However, the heart injury phenotype between WT and LRP5KO mice was the reason for us to initiate lipidomic and other analyses to elucidate the underlying mechanisms. Thus, starting with functional data appears to align more logically with the progression of our investigation. Additionally, we have heeded the reviewer's recommendations by moving Fig. S1A and S1E to the main figure (**Fig. 1C&1F**) and adding NETosis data for Figure 2A in the revised manuscript (**Fig. S2B**).

We also expanded our lipidomic analysis on neutrophils obtained from both WT and KO mice on normal or EFA-free diets as suggested by the reviewer. Volcano plots were generated to show the results (**Fig.2B&2C**). Notably, neutrophils from KO mice exhibited a significant reduction in PUFA-containing lipids compared to those from WT mice. The EFA-free diet led to substantial decreases preferentially in PUFA-containing lipids within neutrophils without significant alterations in the levels of saturated or monounsaturated FA-containing lipids (**Fig. S2I&S2J**). The complete results of our lipidomic analyses are shown in **Supplementary Table 1&2**.

Concern 2:

The manuscript should begin with the lipidomic data in figure 2B on wards, with a more complete/ in-depth analysis. To start with perhaps volcano plots would be better to show the overall lipid species that are changing between the WT and KO neutrophils.

Response: As explained in our response to the 1st question, it appears to be more logical not to start with the lipidomics data. In addition, we generated volcano plots to show the overall lipid species that are changing between the WT and KO (please see **Fig. 2B&C**)

Concern 3:

It would be important to show the expression levels of LRP5 across all these cells types by western blot. The conclusion to the end of the figure in the results section should be removed or re-worded as it shows myeloid/innate cells seem to use LRP5 to take up PUFAS, but not lymphoid cells (i.e. T cell). The authors should also include B cells in this analysis.

Response: We followed the reviewer's suggestion and detected the expression levels of LRP5 across all these cell types by western blot. The result is shown in **Fig.S2L** of the revised manuscript. Furthermore, we performed the PUFA uptake assay in B cells from Rosa26-CreER2 LRP5^{ff} mice treated with tamoxifen, together with their corresponding control cells. Our analysis revealed no difference in the uptake of ¹⁴C-EPA and ¹⁴C-ARA between WT and LRP5KO B cells, so we have rephrased the result section per the reviewer's suggestions.

Concern 4:

Figure 6: The data presented here is very interesting. How were the neutrophil lipidomes altered in the WT and KO mice between resting vs activated (i.e. in A; WT vs KO in the inflammatory model and/or B; control vs fMLP).

Response: We thank the reviewer for raising this excellent question and agree that it is interesting to know how lipids are altered in the WT and KO mice between resting vs activated neutrophils. To address this question, we compared bone marrow derived neutrophils, typically considered more quiescent, and peritoneal neutrophils, which were recruited to peritonea by inflammatory stimulus thioglycolate and are recognized as quasi-activated cells, using targeted LC-MS analysis of a number of PUFA-containing lipids identified by our lipidomic analysis. While LRP5-KO caused reductions in the tested PUFA-containing phospholipids in both bone marrow-derived and peritoneal neutrophils compared to their corresponding WT cells, we also observed apparent increases in PUFA-containing lipids in the peritoneal neutrophils compared to BM neutrophils. By contrast, the phospholipids containing mono/unsaturated fatty acids, which were used as controls, showed no differences not only between WT and KO but also between bone marrow and peritoneal neutrophils. Because we intend to further investigate the mechanisms underlying the unexpected difference in PUFA-containing lipids between peritoneal and BM neutrophils, we thus present the data only to the reviewer (**Fig. R1**).

Responses to the comments of reviewer #3

Overall Comment:

The authors did a great deal of work with innovative findings presented.

Response: We thank the reviewer for considering our study innovative.

Concern 1:

The authors described in the lipidomics method section that both shotgun lipidomics and targeted LC-MS/MS lipidomics were performed. With MRM transitions described for only 8 phospholipids, I assumed that these 8 phospholipids were measured by targeted LC-MS/MS and the rest of the features which were listed in the Supplementary Table 1 were measured by shotgun lipidomics. Please clarify which lipid targets were measured by which platform since it is very confusing. In addition, please provide analytical information about sample extraction for shotgun lipidomics or refer to a method paper if available.

Response: We thank the reviewer for pointing out that we didn't clarify the lipid analysis platform adequately. To address this concern, we clarified which platforms were used in the manuscript and we added information and cited literature about sample extraction for shotgun lipidomics in the Method.

Concern 2:

For targeted LC-MS/MS analysis, the cell count ranged from 3-10 million which varied among different biological samples, how was sample-based normalization performed?

Response: Considering the differences in abundance among various phospholipids, we employed varying cell quantities for our measurements, but the final data has been normalized to the cell number. In **Fig. S2E**, the bar charts were presented as pmol per million cells. We have included this information in the Method.

Concern 3:

The authors should indicate how quality control was performed for lipidomic analyses including batch design, the use of a pooled sample for quality control and data pre-processing strategies. Were biological samples randomized for LC-MS analysis? Which correction strategy was performed for isotopic interferences?

Response: We appreciate the reviewer's comments. Our shotgun lipidomic analysis was performed at Lipotype GmbH (Dresden, Germany). According to the report they provided, lipids were extracted using analytical-grade chloroform and methanol. Samples were spiked with lipid class-specific internal standards prior to extraction. After drying and re-suspending in MS acquisition mixture, lipid extracts were subjected to mass spectrometric analysis. For spectra acquisition, mass spectra were acquired on a hybrid quadrupole/Orbitrap mass spectrometer equipped with an automated nano-flow electrospray ion source in both positive and negative ion

modes. For data processing and normalization, lipid identification was performed using LipotypeXplorer with unprocessed (*.raw format) mass spectra. For the MS-only mode, lipid identification was based on the molecular masses of the intact molecules. The MS-MS mode included the collision-induced fragmentation of lipid molecules, and lipid identification was based on both the intact masses and the masses of the fragments. Prior to normalization and statistical analysis, lipid identifications were filtered according to mass accuracy, occupation threshold, noise, and background. Lists of identified lipids and their intensities were stored in a database optimized for the particular structure inherent to lipidomic datasets. The intensity of lipid class-specific internal standards was used for lipid quantification. For data quality control, the dynamic range for cell culture samples was determined prior to analysis. Based on these data, limits of quantification and coefficients of variation for the different lipid classes were determined. Limits of quantification are in the lower μM to sub- μM range, depending on the lipid class. The average coefficient of variation for a complete set of quantified lipid classes is around 10-15%. Each analysis was accompanied by a set of blank samples to control for a background and a set of quality control reference samples to control for intra-run reproducibility and sample specific issues.

Regarding the question: *Were biological samples randomized for LC-MS analysis?*

For shotgun lipidomics, where a direct infusion method was utilized, there is no liquid chromatography (LC) step. The samples were distributed on the 96-well plate and blinded to the operators at the company. Blanks and reference standard samples were distributed across the plate to control for eventual carry-over effects. For targeted LC-MS analysis, samples were blinded for data collection.

Regarding the question: *Which correction strategy was performed for isotopic interferences?*

Lipids normally undergo isotopic distribution dictated mostly by C13, as other elements can be ignored because their isotopic versions are not so abundant. In order to achieve a proper quantification in our analysis lipids undergo isotopic correction type I and II, meaning that intensities of isotopic peaks of a given lipid molecule are summed up and overlaps between isomers and monoisotopic peaks are corrected.

Concern 4:

In the paragraphs describing Targeted LC-MS/MS analysis of phospholipids and Targeted LC-MS/MS analysis of deuterated EPA, please clarify the exact volume of organic layer transferred and add column size. If available, please refer to a lipidomics method paper.

Response: We followed the reviewer's suggestion and clarified the exact volume of organic layer transferred and column size in the Method section.

Concern 5:

Please carefully check lipid nomenclature in supplementary Table 1. Consider using Lipid Maps nomenclature <https://www.lipidmaps.org/> It is not clear that what 0 following the semicolon means. Define the data unit (or unit-free) for each feature.

Response: We thank the reviewer's comments. The supplementary table 1 was generated by the Lipotype GmbH (Dresden, Germany) and the 0 following the semicolon means the number of hydroxyl groups in that lipid. Following the nomenclature rules on https://www.lipidmaps.org/resources/lipid_web?page=lipids/basics/Nomen/index.htm, we corrected the nomenclature of each lipid in the Table and added detailed nomenclature rules in the Method part.

References

1. Sarparast, M., et al., *Cytochrome P450 Metabolism of Polyunsaturated Fatty Acids and Neurodegeneration*. *Nutrients*, 2020. **12**(11).
2. Gross, E.R., et al., *Cytochrome P450 omega-hydroxylase inhibition reduces infarct size during reperfusion via the sarcolemmal KATP channel*. *J Mol Cell Cardiol*, 2004. **37**(6): p. 1245-9.
3. Balogun, K.A., et al., *Dietary omega-3 polyunsaturated fatty acids alter the fatty acid composition of hepatic and plasma bioactive lipids in C57BL/6 mice: a lipidomic approach*. *PLoS One*, 2013. **8**(11): p. e82399.
4. Oh, D.Y., et al., *GPR120 is an omega-3 fatty acid receptor mediating potent anti-inflammatory and insulin-sensitizing effects*. *Cell*, 2010. **142**(5): p. 687-98.
5. Williams-Bey, Y., et al., *Omega-3 free fatty acids suppress macrophage inflammasome activation by inhibiting NF-kappaB activation and enhancing autophagy*. *PLoS One*, 2014. **9**(6): p. e97957.

REVIEWERS' COMMENTS

Reviewer #1 (Remarks to the Author):

The authors have provided very good responses to the previous concerns raised. There are no further concerns.

Reviewer #2 (Remarks to the Author):

the authors have addressed my comments.

Reviewer #3 (Remarks to the Author):

The authors have made satisfactory revisions and addressed my original concerns. It seems to me that they have also addressed the comments of other reviewers adequately. Now I only have two minor points based on the revised manuscript and I would like to confirm with the authors:

1. In the revised manuscript, the authors detailed the lipid extraction process, stating the use of analytical grade chloroform and methanol (line 554). However, it is common practice in MS-based lipidomics analysis to use MS grade chemicals. Therefore, I kindly request the authors to double-check this aspect.
2. Blank samples were used to monitor background levels, while quality control reference samples were employed to ensure intra-run reproducibility. Could the authors specify the criteria used for assessing the background and quality control samples? Specifying this information for readers would help the overall understanding of the approach of quality check for lipidomics data.

POINT-BY-POINT RESPONSES TO REVIEWERS' COMMENTS:

We sincerely thank all three reviewers for their favorable comments on our initial revisions. Additionally, we deeply appreciate the constructive feedback they offered, which has allowed us to enhance the quality of our manuscript even further.

Responses to the comments of reviewer #1

Comment:

The authors have provided very good responses to the previous concerns raised. There are no further concerns.

Response: We are grateful that the reviewer finds our revision satisfactory.

Responses to the comments of reviewer #2

Comment:

the authors have addressed my comments.

Response: We are grateful for the reviewer's approval of our revision.

Responses to the comments of reviewer #3

Overall comment:

The authors have made satisfactory revisions and addressed my original concerns. It seems to me that they have also addressed the comments of other reviewers adequately. Now I only have two minor points based on the revised manuscript and I would like to confirm with the authors:

Response: We are thankful that the reviewer is satisfied with the majority of our revision. We also appreciate the two minor points provided by the reviewer.

Minor point 1:

In the revised manuscript, the authors detailed the lipid extraction process, stating the use of analytical grade chloroform and methanol (line 554). However, it is common practice in MS-based lipidomics analysis to use MS grade chemicals. Therefore, I kindly request the authors to double-check this aspect.

Response: We really value the reviewer's attention to this minor detail. We double checked with the company who conducted the lipidomic analysis for us and confirmed their use of MS grade chemicals. So, we have updated our method part accordingly (see line 549 in the revised manuscript).

Minor point 2:

Blank samples were used to monitor background levels, while quality control reference samples were employed to ensure intra-run reproducibility. Could the authors specify the criteria used for assessing the background and quality control samples? Specifying this information for readers would help the overall understanding of the approach of quality check for lipidomics data.

Response: We greatly appreciate the reviewer's suggestions. Following the advice, we reached out to Lipotype, the company responsible for the lipidomic analysis. Below are their criteria used for assessing the background and quality control samples, which have been incorporated into the method part (refer to lines 563-566 and lines 571-574 in the revised manuscript).

"All the extractions and sample processing must be done without any incidents. The ionization quality must be satisfactory and no issues were encountered. The internal standard signal-to-noise ratio was in range of hundreds and more, indicating high quality of spectra. The technical reproducibility, as assessed by quality control reference samples (mammalian full blood) included in the same analytical run was very good, and blank samples included in each analytical batch (internal background control) did not contain any unusual background."